



# Uncertainty in water transit time estimation with StorAge Selection functions and tracer data interpolation

Arianna Borriero[1], Rohini Kumar[2], Tam V. Nguyen[1], Jan H. Fleckenstein[1,3], and Stefanie R. Lutz[4]

[1]Department of Hydrogeology, Helmholtz-Centre for Environmental Research - UFZ, Leipzig, Germany
[2]Department of Computational Hydrosystems, Helmholtz-Centre for Environmental Research - UFZ, Leipzig, Germany
[3]Bayreuth Centre of Ecology and Environmental Research, University of Bayreuth, Bayreuth, Germany
[4]Copernicus Institute of Sustainable Development, Department of Environmental Sciences, Utrecht University, Utrecht, the Netherlands

**Correspondence:** Arianna Borriero (arianna.borriero@ufz.de)

**Abstract.** Transit time distributions (TTDs) of streamflow are useful descriptors for understanding flow and solute transport in catchments. Catchment-scale TTDs can be modeled using tracer data (e.g., $\delta^{18}O$; oxygen isotopes) in inflow and outflows, with StorAge Selection (SAS) functions. However, tracer data are often sparse in space and time, so they can be interpolated to increase their spatio-temporal resolution. Also, SAS functions can be parameterized with different forms, but there is no general

agreement on which one should be used. Both of these aspects induce uncertainty in the simulated TTDs, and the individual uncertainty sources as well as their combined effect have not been fully investigated. This study provides a comprehensive analysis of the TTD uncertainty resulting from twelve model setups obtained by combining different interpolation schemes for $\delta^{18}O$ in precipitation, and distinct SAS functions. Furthermore, we evaluated the value of the young water fraction ($F_{yw}$) as an additional constraint for the TTD uncertainty. For each model setup, we found behavioral solutions with satisfactory

model performances for instream $\delta^{18}O$ (Kling-Gupta Efficiency, KGE>0.57). Differences in KGE values were statistically significant, thus showing the relevance of the chosen setup for simulating TTDs. We found a large uncertainty in the simulated TTDs, with a 90% confidence interval varying between 286 and 895 days across all tested setups. Uncertainty in TTDs was mainly associated with the temporal interpolation of $\delta^{18}O$ in precipitation, time-variant SAS function and low flow conditions. The use of $F_{yw}$ as an additional constraint substantially reduced the uncertainty in the predicted TTDs by up to 49%. We

discussed the implications of these results with respect to the study area and the SAS framework, in order to identify ways to improve uncertainty characterization and water age simulations in TTD-based models.

## 1   Introduction

Understanding how catchments store and release water of different ages has significant implications for flow and solute transport as water ages encapsulate information about flowpaths characteristics (McGuire and McDonnel, 2006; Botter et al., 2011),

contact time of solutes with the soil matrix (Benettin et al., 2015a; Hrachowitz et al., 2016), and vulnerability assessment (Kumar et al., 2020). This plays an important role for water resources protection and management, as well as requires a tool that can effectively describe catchment-scale transport processes (Rinaldo and Marani, 1987). The water age in outflows is commonly



referred to as transit time (TT), i.e., the time elapsed between the entry of a water parcel into the catchment via precipitation and its exit via streamflow or evapotranspiration. Accordingly, the transit time distribution (TTD) describes the whole spectrum of the transit times in outflows (Botter et al., 2005; Van der Velde et al., 2010). Early studies have often assumed simplified steady-state transport models, resulting in time-invariant TTDs (Niemi, 1977; Rinaldo et al., 2006). However, experimental simulations showed that TTDs are time-variant due to variability in meteorological forcing (Botter et al., 2010; Hrachowitz et al., 2010; Heidbüchel et al., 2020). A promising tool for representing time-variant TTDs are StorAge Selection (SAS) functions, which describe how catchments selectively remove water of different ages from storage to outflows (Rinaldo et al., 2015; Harman, 2019). SAS functions have led to a new framework of non-stationary transport models based on water age, which have been successfully applied in various transport modelling studies (Benettin et al., 2015b; Queloz et al., 2015; Kim et al., 2016; Lutz et al., 2017; Wilusz et al., 2017; Nguyen et al., 2021).

Model-based TTDs are subjected to uncertainty which limits their ability for decision support. In general, model prediction uncertainty stems from model inputs, structure, and parameters (Beven and Freer, 2001). As TTDs are not directly observable, conservative environmental tracers (e.g., $\delta^{18}O$; oxygen isotopes) in inflow and outflows are commonly used to infer water ages (Hrachowitz et al., 2013; Birkel and Soulsby, 2015; Stockinger et al., 2015). Long-term, high-frequency tracer data with an appropriate spatial distribution are generally recommended for sufficient understanding of the TTDs dynamics across a wide range of fast and heterogeneous hydrological behaviors (Kirchner et al., 2004; Danesh-Yazdi et al., 2016; von Freyberg et al., 2017). Therefore, the lack of an appropriate tracer data coverage can prevent our understanding of the TTDs dynamics at the desired resolution (McGuire and McDonnel, 2006). Further uncertainty emerges from the model structure due to the difficulty in representing physical processes because of our incomplete knowledge of complex reality (Ajami et al., 2007). Finally, specification of model parameters is also an important source of uncertainty (Beven, 2006; Kirchner, 2006), as the best-fit parameters may suffer from equifinality (Schoups et al., 2008).

A few studies have investigated the uncertainty in the estimated TTDs with SAS models. Danesh-Yazdi et al. (2018) and Jing et al. (2019) have analysed the effect of the interactions between distinct flow domains, external forcing and recharge rate on TTDs. Several works (Benettin et al., 2017; Wilusz et al., 2017; Rodriguez et al., 2018, 2021) have explored model parameter uncertainty, and suggested that additional types of tracers, data on physical characteristics of the catchment, and parsimonious parameterization may help to further reduce parameters uncertainty. More recently, Buzacott et al. (2020) investigated how gap-filling of the $\delta^{18}O$ record in precipitation propagated uncertainty into the simulated mean water transit time (MTT), i.e., the average time it takes for water to leave the catchment (McDonnel et al., 2010).

Despite the studies cited above, there are other aspects causing uncertainty in the simulated TTDs, which have not yet been fully addressed. First, isotope data are generally sparse globally in space and time (von Freyberg et al., 2022), due to laborious and costly sampling campaigns limited to well-equipped areas (Tetzlaff et al., 2018). For this reason, spatio-temporal interpolation of the isotope composition in precipitation can be used to get greater data coverage, required for modelling purposes. Second, SAS functions, employed to model TTDs, must be parameterized, and commonly used parameterizations are the power law (Benettin et al., 2017; Asadollahi et al., 2020) and beta distribution (van der Velde et al., 2012; Drever and Hrachowitz, 2017). However, there is no general agreement on which SAS function should be used as the hydrological





processes which control subsurface mixing, hence TTDs dynamics, are distinct across different catchments. Therefore, the
most convenient approach is to simply rely on one parameterization over another, and estimate its parameters (Harman, 2015).
Both of these aspects induce model input, structure and parameter uncertainty in the simulated TTDs. To date, the individual
uncertainty sources and their combined effect on the modeled TTDs have not been adequately discussed.

To reduce the estimated TTD uncertainty, various studies have suggested combining stable and decaying tracers (Duvert
et al., 2016), the latter best imparting old water ages. Others have proposed high sampling resolution (e.g., daily) to provide
better insights into short-term events (Timbe et al., 2015). Nonetheless, when long-term, high frequency tracer data are scarce,
methods that rely on short-term, low frequency data can become an alternative option to reduce TTD uncertainty. One such
example is the young water fraction ($F_{yw}$; streamflow ratio younger than 2-3 months (Kirchner, 2016a)), which can be easily
retrieved from low frequency tracer data covering relatively short period. The estimation of $F_{yw}$ is based on the amplitude ratio
of the seasonal cycles in stable water isotopes in precipitation and streamflow. $F_{yw}$ represents an alternative descriptor for TTD,
a robust metric under both spatially heterogeneous and non-stationary conditions, as well as less prone to a large aggregation
bias than MTTs (Kirchner, 2016b). Benettin et al. (2017) proposed to use $F_{yw}$ to restrict model parameter values, while Lutz
et al. (2018) implemented it as an additional constraint in the calibration of gamma-distributed TTDs. However, to the best of
our knowledge, no studies have attempted to employ $F_{yw}$ for constraining predictive uncertainties in SAS-based TTDs.

This study bridges the aforementioned gaps by specifically exploring the combined effect of sparse input tracer data and
model parameterizations on the simulated TTDs. We investigated TTD uncertainty using a SAS-based catchment-scale trans-
port model applied to the Upper Selke catchment, Germany. We evaluated TTDs resulting from twelve model setups obtained
by combining distinct interpolation techniques of $\delta^{18}O$ in precipitation, and parameterizations of SAS functions. For each
model setup, we searched for behavioral parameter sets (i.e., those providing acceptable predictions) based on model perfor-
mance for instream $\delta^{18}O$. Afterwards, we evaluated the sources of uncertainty, as well as their combined effects, in the resulting
TTDs and, finally, we tested whether and to what extent $F_{yw}$ could provide valuable information to further constrain the un-
certainty. Overall, our results provide new insights into the uncertainty characterization of TTDs, particularly in the absence of
high-frequency tracer data, and the use of SAS functions.

## 2  Study area and data

The Upper Selke catchment is located in the Harz Mountains in Saxony-Anhalt, central Germany (Fig. 1). The study site
is part of the Bode region, an intensively monitored area within the TERENO (TERrestrial ENvironmental Observatories;
Wollschläger et al., 2017) network. The catchment has a drainage area of 184 km$^2$, the altitude ranges between 184 and 594 m
above mean sea level, and the mean slope is 7.65%. Land use is dominated by forest (broadleaf, coniferous and mixed forest)
and agricultural land (winter cereals, rapeseed and maize), representing 72% and 21% of the catchment, respectively. The soil
is largely composed of cambisols and the underlying geology consists of schist and claystone, resulting in a predominance of
relatively shallow flowpaths (Dupas et al., 2017; Yang, J. et al., 2018).





Daily hydroclimatic and monthly tracer data in the Upper Selke were available for the period between February 2013 and May 2015. Precipitation (P) was taken from the German weather service, while discharge (Q) and evapotranspiration (ET) were simulated data obtained from the mesoscale Hydrological Model (mHM; (Samaniego et al., 2010; Kumar et al., 2013)) since continuous measurements were not available for the given outlet and period. A thorough evaluation of mHM performance for past measurements have been conducted in previous studies (Zink et al., 2017; Yang, X. et al., 2018; Nguyen et al., 2021). The

average annual P, Q and ET are 703, 108, 596 mm, respectively. The area is characterized by high flow during November-May (average Q = 0.88 m$^3$/s) and low flow during June-October (average Q = 0.42 m$^3$/s). Evapotranspiration is higher in June (109 mm/month) and lower in December (10 mm/month). The average monthly temperature ranges from -0.7°C in January to 17°C in July. The $\delta^{18}$O values in precipitation ($\delta^{18}$O$_P$) and in streamflow ($\delta^{18}$O$_Q$) at monthly resolution were taken from Lutz et al. (2018) (Fig. S1). $\delta^{18}$O$_P$ were used in the form of "raw" (i.e., values collected at the catchment outlet) and processed (i.e.,

spatially interpolated using kriging) data (Section 3.2). The variability in $\delta^{18}$O$_P$ was larger than $\delta^{18}$O$_Q$ because of the damped precipitation signal due to mixing and dispersion within the catchment. Temperature dependence caused more depleted (i.e., more negative) $\delta^{18}$O$_P$ in winter than in summer (Fig. S1).

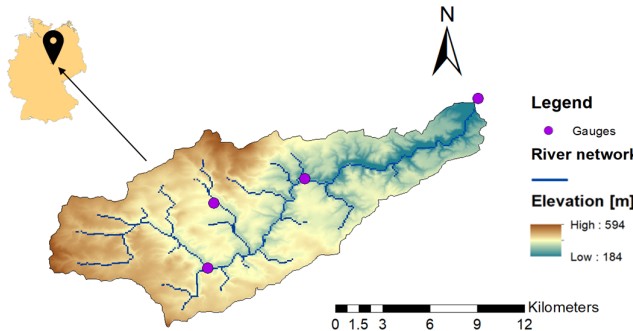

**Figure 1.** Upper Selke catchment with precipitation sampling points (purple dots), river network (blue lines), and elevation in meters above sea level as colored map; location of the Upper Selke catchment in Germany (upper left corner).

## 3   Methods

### 3.1   Catchment-scale transport model

In this study, we used the *tran-SAS* model (Benettin and Bertuzzo, 2018) for describing the catchment-scale water mixing and solute transport based on SAS functions. The catchment was conceptualized as a single storage *S(t)* (mm), whose water-age balance can be expressed as follows (Benettin and Bertuzzo, 2018):

$$S(t) = S_0 + V(t) \tag{1}$$





$$\frac{\partial S_T(T,t)}{\partial t} + \frac{\partial S_T(T,t)}{\partial T} = P(t) - Q(t) \cdot \Omega_Q(S_T,t) - ET(t) \cdot \Omega_{ET}(S_T,t) \qquad (2)$$

Initial condition: $S_\mathrm{T}(T, t=0) = S_{\mathrm{T}_0}$     (3)

Boundary condition: $S_\mathrm{T}(0,t) = 0$     (4)

where $S_0$ (mm) is the initial storage, $V(t)$ (mm) are the storage variations, $P(t)$ (mm/d), $Q(t)$ (mm/d), and $ET(t)$ (mm/d) are precipitation, discharge and evapotranspiration, respectively, $S_T(T,t)$ (mm) is the age-ranked storage, $S_{T0}$ (mm) is the initial
age-ranked storage, and $\Omega_Q(S_T,t)$ (-) and $\Omega_{ET}(S_T,t)$ (-) are the cumulative SAS functions for $Q$ and $ET$, respectively.

By definition, the TTD of streamflow $p_Q(T,t)$ (d$^{-1}$) is calculated as follows (Benettin and Bertuzzo, 2018):

$$p_Q(T,t) = \frac{\partial \Omega_Q(S_T,t)}{\partial S_T} \cdot \frac{\partial S_T}{\partial T}. \qquad (5)$$

The isotopic signature in streamflow $C_Q(t)$ (‰) can be obtained from (Benettin and Bertuzzo, 2018):

$$C_Q(t) = \int\limits_{0}^{+\infty} C_S(T,t) \cdot p_Q(T,t) \cdot dT \qquad (6)$$

where $C_S(T,t)$ (‰) is the isotopic signature of a water parcel in storage. Equations 5 and 6 also apply for $ET$.

In this study, we tested three SAS parameterizations: the power law time-invariant (PLTI; Eq. 7 (Queloz et al., 2015)), power law time-variant (PLTV; Eq. 8 (Benettin et al., 2017)), and beta distribution (BETA; Eq. 9 (Drever and Hrachowitz, 2017)). They can be expressed as probability density functions in terms of the normalized age-ranked storage $P_S(T,t)$ (-):

$$\omega(P_S(T,t),t) = k \cdot (P_S(T,t))^{k-1} \qquad (7)$$

$$\omega(P_S(T,t),t) = k(t) \cdot (P_S(T,t))^{k(t)-1} \qquad (8)$$

$$\omega(P_S(T,t),t) = \frac{(P_S(T,t))^{\alpha-1} \cdot (1 - P_S(T,t))^{\beta-1}}{B(\alpha,\beta)}. \qquad (9)$$

The parameters $k$, $\alpha$ and $\beta$ determine the catchment's water age preference for outflow, while $B(\alpha,\beta)$ is the two-parameter beta function. If $k<1$, or if $\alpha<1$ and $\beta>1$, the system tends to discharge young water. If $k>1$, or if $\alpha>1$ and $\beta<1$, the catchment preferably releases old water. The case of $k=1$ or $\alpha=\beta=1$ describes no selection preference (i.e., complete water mixing). PLTV
is characterized by $k(t)$ varying linearly over time between two extremes $k_1$ and $k_2$ as a function of the catchment wetness $wi$ (-), i.e., $wi(t) = (S(t)-S_{min})/(S_{max}-S_{min})$, where $S_{min}$ and $S_{max}$ are the minimum and maximum storage values over the entire period.

## 3.2 Interpolation techniques for $\delta^{18}$O in precipitation

We tested the model with two spatial and two temporal interpolation methods of tracer data to explore the TTD uncertainty resulting from model input. To evaluate the effect of the spatial interpolation, we first set a base case using monthly raw
$\delta^{18}$O$_P$ taken from Lutz et al. (2018), corresponding to the values collected at the catchment outlet (i.e., Meisdorf station).





Second, we used the spatially interpolated $\delta^{18}O_P$ estimates from Lutz et al. (2018), which are based on raw observations from 24 precipitation collectors spread over the larger area of the Bode region. The spatial interpolation in Lutz et al. (2018) was conducted using kriging with altitude as an external drift. The kriged $\delta^{18}O_P$ were further weighted with spatially distributed monthly precipitation to obtain representative estimates for the study region.

SAS model results are sensitive to the choice of the temporal resolution of input tracer data, and shorter time steps are generally recommended to achieve a satisfactory level of detail (Benettin and Bertuzzo, 2018). Additionally, a forward Euler scheme was employed to solve Eq. 2, whose precision increases with high frequency time steps. For this reason, we reconstructed daily $\delta^{18}O_P$ estimates from monthly values with two different interpolation schemes. First, we used a step function in which the values between two consecutive samples assumed the value of the last sample. Second, we used a sine interpolation based on

the assumption that the $\delta^{18}O_P$ values follow a seasonal cycle (Fig. S1 in the Supplement, (Feng et al., 2009)), whose signature over a period of one year can be described by (Kirchner, 2016a):

$$\delta^{18}O_P(t) = a_P \cdot cos(2 \cdot \pi \cdot f \cdot t) + b_P \cdot sin(2 \cdot \pi \cdot f \cdot t) + k_P \tag{10}$$

where $a$ and $b$ are regression coefficients, $t$ is the time (decimal years), $f$ is the frequency (yr$^{-1}$) and $k$ (‰) is the vertical offset of the isotope signal. The coefficients $a$ and $b$ were estimated by fitting Eq. 10 to monthly $\delta^{18}O_P$ values using the iteratively

re-weighted least squares (IRLS) estimation (von Freyberg et al., 2018). Subsequently, the estimated regression coefficients were used in Eq. 10 to obtain isotope data at daily frequency. Figure S2 in the Supplement displays the simulated kriged and raw $\delta^{18}O_P$ values via step function and sine interpolation.

### 3.3   Estimation of the young water fraction ($F_{yw}$)

$F_{yw}$ was estimated through a sine-wave fit, which assumes that the seasonal cycle in $\delta^{18}O_P$ is transmitted into $\delta^{18}O_Q$ as a

damped and phase-shifted signal (Soulsby et al., 2006). In line with (Bliss, 1970), the isotopic signal for $\delta^{18}O_P$ and $\delta^{18}O_Q$ over a year can be written respectively as (Kirchner, 2016a):

$$c_P(t) = A_P \cdot sin(2 \cdot \pi \cdot f \cdot t - \phi_P) + k_P \tag{11}$$

$$c_Q(t) = A_Q \cdot sin(2 \cdot \pi \cdot f \cdot t - \phi_Q) + k_Q \tag{12}$$

where $A$ is the tracer cycle amplitude (‰), $\phi$ is the phase of the cycle (radians), $t$ is the time (decimal years), $f$ is the frequency

(yr$^{-1}$) and $k$ (‰) is the vertical offset of the isotope signal. The amplitude ratio defines the young water fraction $F_{yw}$ (Kirchner, 2016a):

$$F_{yw}^{est} = \frac{A_q}{A_p}. \tag{13}$$

$A_P$ and $A_Q$ were calculated with the IRLS method by fitting Eqs. 11 and 12 to monthly $\delta^{18}O_P$ and $\delta^{18}O_Q$ samples volume-weighted with the corresponding $P$ and $Q$ rates; thus, the flow-weighted $F_{yw}^{est}$ was computed (von Freyberg et al., 2018).

Gaussian error propagation was applied to assess the standard error ($\pm$ SE) of $F_{yw}^{est}$ as an uncertainty measure.





This tracer-based $F_{yw}^{est}$ was compared with the simulated young water fraction $F_{yw}^{sim}$ obtained via the SAS model. To determine $F_{yw}^{sim}$, we averaged the TTDs computed at any time step; then, we flow-weighted the averaged TTDs to obtain the long-term TTD over the entire study period, also known as marginal TTD (Heidbüchel et al., 2012). From the marginal TTD, we computed $F_{yw}^{sim}$ as:

$$F_{yw}^{sim} = P_Q(T = \tau_{yw}). \tag{14}$$

where $\tau_{yw}$ is the threshold age set to 75 days, which falls within 2-3 months defining $F_{yw}$ (Kirchner, 2016a). We used $F_{yw}^{est} \pm SE$ instead of a specific value to account for the uncertainty associated with a fixed threshold age of $\tau_{yw}=75$ days for $F_{yw}^{sim}$.

### 3.4 Experimental design

In this study, different scenarios were used to quantify uncertainty in the modeled results. We tested twelve setups composed of three SAS functions (PLTI, PLTV, BETA), two temporal (step and sine function) and two spatial (raw and kriging values) interpolation techniques (Table 1). For each setup, we performed a Monte Carlo experiment by running the model with 10,000 parameter sets generated by the Latin Hypercube Sampling (LHS, McKay et al., 1979). Model parameters and their search ranges are shown in Table 2.

**Table 1.** List of model setups.

| setup | interpolation | SAS function |
|---|---|---|
| 1 | step function kriged $\delta^{18}O_P$ | PLTI |
| 2 | | PLTV |
| 3 | | BETA |
| 4 | step function raw $\delta^{18}O_P$ | PLTI |
| 5 | | PLTV |
| 6 | | BETA |
| 7 | sine function kriged $\delta^{18}O_P$ | PLTI |
| 8 | | PLTV |
| 9 | | BETA |
| 10 | sine function raw $\delta^{18}O_P$ | PLTI |
| 11 | | PLTV |
| 12 | | BETA |

**Table 2.** Model parameters and search ranges.

| SAS parameter | Symbol | Unit | Lower Bound | Upper Bound |
|---|---|---|---|---|
| Discharge SAS parameter | $k_Q$ | [-] | 0.1 | 2 |
| Discharge SAS parameter | $k_{Q1}$ or $\alpha$ | [-] | 0.1 | 2 |
| Discharge SAS parameter | $k_{Q2}$ or $\beta$ | [-] | 0.1 | 2 |
| Evapotranspiration SAS parameter | $k_{ET}$ | [-] | 0.1 | 2 |
| Initial storage | $S_0$ | [mm] | 300 | 3000 |





A 5 years warm-up period (i.e., repetition of the input data) from February 2008 to January 2013 was performed to reduce the
impact of the model initialization. The period from February 2013 to May 2015 was used to infer behavioral model parameters
(i.e., parameter sets giving acceptable predictions), and subsequently to interpret the model results. The initial concentration of
$\delta^{18}$O in storage was set to 9.2 ‰ coinciding with the mean $\delta^{18}O_Q$ over the study period.

The generalized likelihood uncertainty estimation (GLUE Beven and Binley, 1992) was applied to determine the behavioral
parameter sets in terms of the Kling-Gupta efficiency (KGE Gupta et al., 2009) for observed and simulated $\delta^{18}O_Q$ values. The
best 5% parameter sets were selected as behavioral, from which we constructed the 90% confidence intervals (CI) to refine
limits of the behavioral solutions for every output variable. Instead of fixing a threshold limit based on KGE, we set the sample
size to allow for comparability of the results across different model setups.

From the daily TTDs, we extracted the temporal evolution of the daily median transit time ($TT_{50}$ (days), i.e., the time elapsed
until 50% of the infiltrated water is transferred to the outflow (Sprenger et al., 2016)), and used it as a metric for the streamflow
age. This was done because TTDs are typically skewed with long tails (Kirchner et al., 2001); hence, the median is often more
suitable as it is less impacted by the poor identifiability of the older water components (Benettin et al., 2017).

In a final step, the behavioral parameter sets identified via GLUE were further constrained with $F_{yw}^{est}$; the behavioral solu-
tions providing $F_{yw}^{sim}$ (Eq. 14) within the range $F_{yw}^{est} \pm$ SE (Eq. 13) were chosen as the final behavioral solutions.

## 4   Results

### 4.1   Simulated $\delta^{18}$O in streamflow and model performances

Modeled $\delta^{18}$O in streamflow ($\delta^{18}O_Q$) represented by the 90% confidence interval (CI) in the ensemble solution are displayed
in Fig. 2. The results reveal how the predicted $\delta^{18}O_Q$ values enveloped the measured isotopic signature by reproducing its
seasonal fluctuations, with depleted (i.e., more negative) values in winter and enriched (i.e., more positive) values in summer.
However, the second half of the study period is characterized by more enriched predicted $\delta^{18}O_Q$ values than the measured ones.
Although the behavioral parameter sets are able to capture the seasonal isotopic trend, they poorly reproduced the exact values;
therefore the ensemble simulations are characterized by a non-negligible uncertainty.

Figure 2 shows the distinct effects of the interpolated input tracer data and model parameterization on the simulated $\delta^{18}O_Q$
values. The step function interpolation generated an erratic isotopic signature in streamflow with flashy fluctuations, explicitly
visible in Fig. 2c and f. On the other hand, the sine interpolation of $\delta^{18}O_P$ values yielded a smooth response in the simulated
$\delta^{18}O_Q$ values (Fig. 2g-l). Conversely, no clear visual difference was found between the raw (Fig. 2d-f and j-l) and kriged (Fig.
2a-c and g-i) $\delta^{18}O_P$ samples as their general patterns match (Fig. 2S in the Supplement). Likewise, distinct SAS parameteriza-
tions produced remarkable differences in the simulated $\delta^{18}O_Q$ values. BETA induced a large 90% CI in the $\delta^{18}O_Q$ values (Fig.
2c, f, i and l), and wide tracer cycle amplitudes. In contrast, both PLTI and PLTV (Fig. 2a, b, d, e, g, h, j and k) produced a
substantially narrow 90% CI with small amplitudes of the simulated $\delta^{18}O_Q$ values.

Despite the differences in the predicted $\delta^{18}O_Q$ values, all simulations can be considered satisfactory given the KGE values
ranging between 0.57 and 0.75, across all tested setups (Fig. 3). These performances can be classified from intermediate





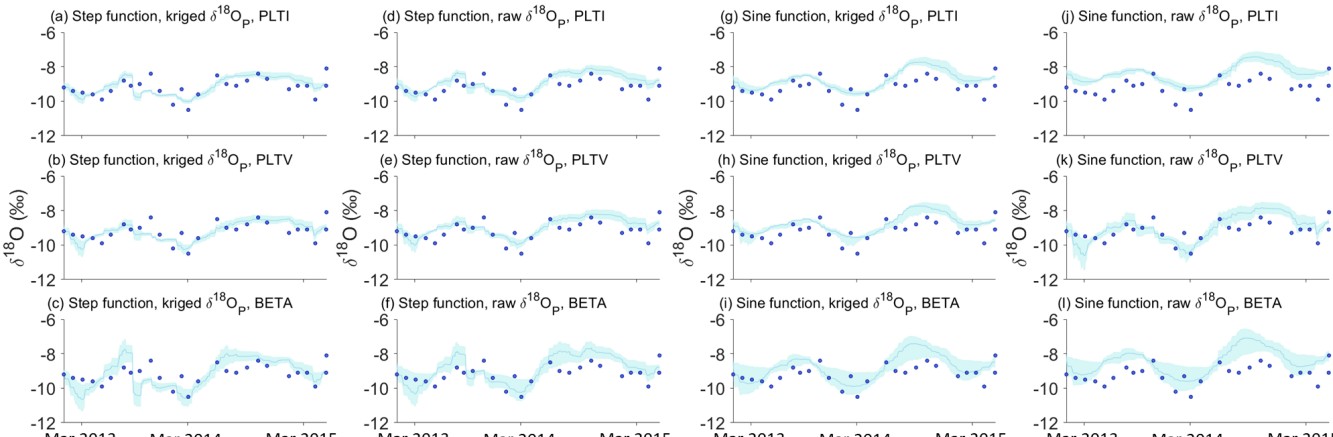

**Figure 2.** Predicted $\delta^{18}$O values in streamflow identified by 5% best KGE. Dark blue filled circles represent the observed data; the light blue line and the shaded area represent, respectively, the ensemble mean of all possible solutions and its variation range according to the 90% CI.

(Thiemig et al., 2013) to good (Andersson et al., 2017; Sutanudjaja et al., 2018). When considering the best fit, the combination of the step function interpolation and raw $\delta^{18}O_P$ values performed best. Additionally, PLTV yielded slightly better KGE values than PLTI and BETA when grouping the setups with the same interpolation techniques of $\delta^{18}O_P$. Differences in the KGE means

were statistically insignificant (t-test with p-values > 0.05) between setups 1 and 3, as well as among setups 6, 7 and 9 (Table 1), and this largely matches our visual analysis. Contrarily, the differences in the remaining setups were statistically significant (p-values < 0.05), indicating that a priori methodological choices (i.e., interpolation techniques of $\delta^{18}O_P$ values and/or SAS parameterization) strongly impact on the overall results. Notwithstanding, this does not mean that we can clearly identify the most suitable setup .

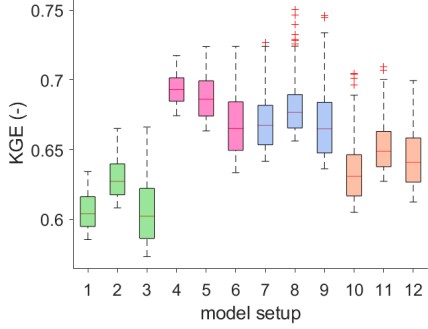

**Figure 3.** Boxplot of model performance ranges in behavioral solutions obtained with 5% best KGE; boxplots filled with the same colors represent model setups characterized by the same interpolation scheme in space and time. On each box, the central mark indicates the median, and the bottom and top edges of the box indicate the 25th and 75th percentiles, respectively. The whiskers extend to the most extreme data points not considered outliers, and the outliers are plotted individually using the '+' mark.





Ranges of the behavioral SAS parameters for the tested setups are summarized in Table S1 in the Supplement. Parameters for $Q$ (i.e., $k_Q$, $k_{Q1}$, $k_{Q2}$, $\alpha$ and $\beta$) were different across the setups although, in general, they were relatively narrow and well identified. However, the behavioral parameters were better constrained when using the step function interpolation since their range was, on average, 34% narrower than that provided by the sine interpolation, across all the SAS parameterizations. The parameters $k_{Q1}$ and $\alpha$ were also better identified than $k_{Q2}$ and $\beta$, since their range was, on average, 22% narrower, in all tested

setups. Conversely, there was no significant difference in the parameters ranges when using kriged or raw $\delta^{18}O_P$ values. The evapotranspiration parameter (i.e., $k_{ET}$) was poorly identified in all setups as any value in the search range provided equally good results. The initial storage $S_0$ was only partially constrained as any value between 400 mm and 2340 mm was considered acceptable.

### 4.2    Simulated transit times and model uncertainty

Figure 4 illustrates the 90% CI of the behavioral solutions for the predicted median transit times ($TT_{50}$). The results show that the model simulates very different ranges of $TT_{50}$ values based on the tested setups. When using PLTI and BETA, the 90% CI was relatively stable with small fluctuations throughout the simulation period, compared to PLTV (Fig. 4a, c, d, f, g, i, j and l). However, minor differences emerged across the simulated $TT_{50}$, as a result of the distinct interpolation techniques used for $\delta^{18}O_P$ values. The 90% CI was on average slightly larger by 11%, across all tested setups, when using kriged $\delta^{18}O_P$ (Fig.

4a-c and g-i) rather than raw $\delta^{18}O_P$ values (Fig. 4d-f and j-l). Also, the use of the sine function generated a 90% CI being on average 22% narrower across all tested setups, (Fig. 4g-l) with respect to the step function (Fig. 4a-f), notably within high flow conditions. In addition, the behavioral solutions obtained with BETA were more skewed towards shorter mean $TT_{50}$ values, across all tested setups, in contrast to PLTI and PLTV (Fig. 4c, f, i and l).

       Behavioral solutions obtained with PLTV revealed a similar pattern regardless of the interpolation employed (Fig. 4b, e, h

and k). Nonetheless, there was a noticeable difference in the 90% CI under distinct flow regimes. During low flows and dry periods (i.e., late summer and autumn), the predicted $TT_{50}$ show large uncertainties ranging at most between 286 and 895 days for the same moment in time (Fig. 4e). Conversely, during high flows (i.e., winter and spring), the 90% CI was narrow and varied at least between 135 and 163 days only (Fig. 4h). The large 90% CI and the notable differences across the tested setups highlight the sensitivity and, in turn, the uncertainty of predicted $TT_{50}$ to the model parameterization, temporal interpolation

of input data and hydrologic conditions. In contrast, the use of raw or kriged $\delta^{18}O_P$ samples produced small differences in the estimated $TT_{50}$; thus the spatial interpolation technique did not substantially affect the water age simulations.

       In general, the variability of the predicted $TT_{50}$ is controlled by the hydrological state of the system (Fig. 4). Discharge events reduced $TT_{50}$ values, while low flow periods were associated with a longer estimated $TT_{50}$. This is expected as streamflow during high (low) flows is dominated by near-surface runoff (groundwater) with shallow (deep) flowpaths leading to a shorter

(longer) $TT_{50}$. Such differences are particularly visible with PLTV (Fig. 4b, e, h , and k) as the exponent $k_Q(t)$ shifts the water selection preference over time as a function of the wet/dry conditions; this makes the variability of $TT_{50}$ more pronounced than that of PLTI and BETA, whose SAS parameters for $Q$ are constant over time.




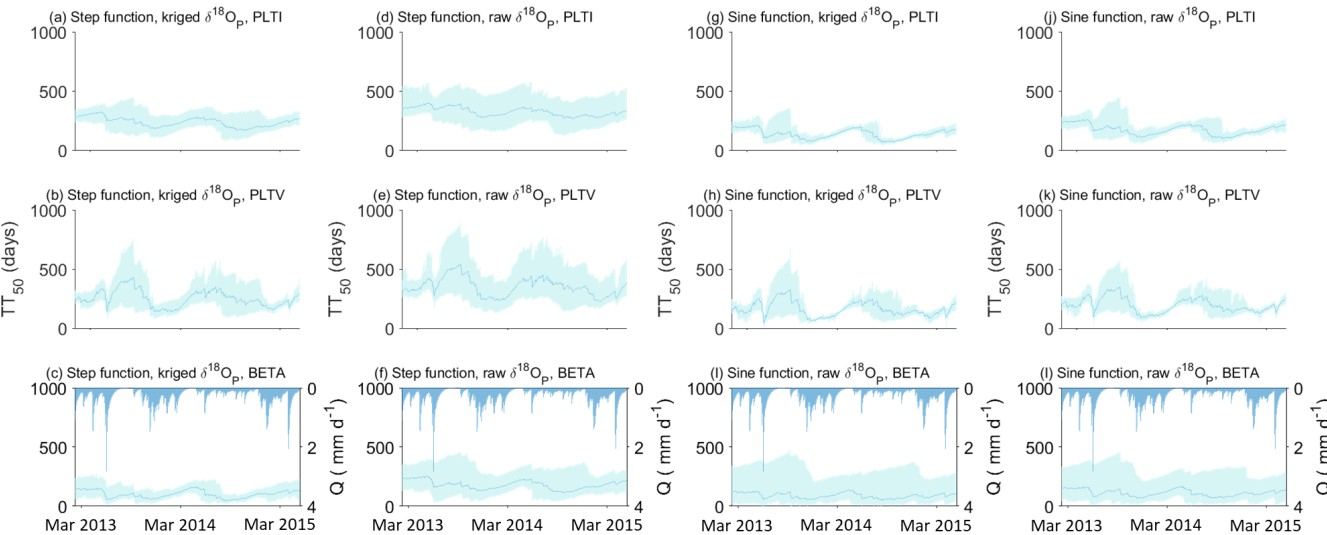

**Figure 4.** Predicted $TT_{50}$ of streamflow identified by 5% best KGE; the light blue line and the shaded area represent, respectively, the ensemble mean of all possible solutions and its variation range according to the 90% CI.

### 4.3 Use of the young water fraction ($F_{yw}$)

The sine-wave fit produced young water fraction ($F_{yw}^{est}$) values equal to 0.25±0.08 and 0.22±0.07 for kriged and raw $\delta^{18}O_P$

samples, respectively. This means that approximately a quarter of the total streamflow is composed of water with ages younger than 2-3 months. Behavioral solutions produced SAS-based $F_{yw}^{sim}$ values ranging between 0.22 and 0.54, across all tested setups, hence being much larger than $F_{yw}^{est}$. There were considerable differences in $F_{yw}^{sim}$ across the SAS parameterizations as BETA produced a 37% larger $F_{yw}^{sim}$ than that obtained with PLTI and PLTV, across the interpolation schemes. Moreover, $F_{yw}^{sim}$ obtained with the sine interpolation was 31% larger than that of the step interpolation, across the tested setups. Conversely, no

difference was found when using kriged or raw $\delta^{18}O_P$ samples.

Figure 5 displays the simulated $TT_{50}$ before (blue) and after (pink) using $F_{yw}^{est}$ to reduce the uncertainty in the simulated $TT_{50}$. The results show that the addition of $F_{yw}^{est}$ resulted in substantial changes in the 90% CI of $TT_{50}$, and the uncertainty reduction, in terms of percentage change, varied between 0% (Fig. 5b, d and e) and 49% (Fig. 5i) across the analysed setups. The lowest reduction occurred with the power law SAS functions and step function interpolation, with an average of 1% across

the tested setups (Fig. 5a, b, d and e). In contrast, the 90% CI was largely reduced with BETA and the sine interpolation with an average of 42% across the setups (Fig. 5c, f, i and l). This clearly demonstrates the value of $F_{yw}^{est}$ as an additional constraint in $TT_{50}$ modelling.

### 4.4 Catchment-scale water release

SAS functions provided valuable insights into the catchment-scale water release dynamics. Figure 6 presents the behavioral

solutions releasing water of different ages identified by 5% best KGE (blue) and after adding $F_{yw}^{est}$ (pink), respectively.





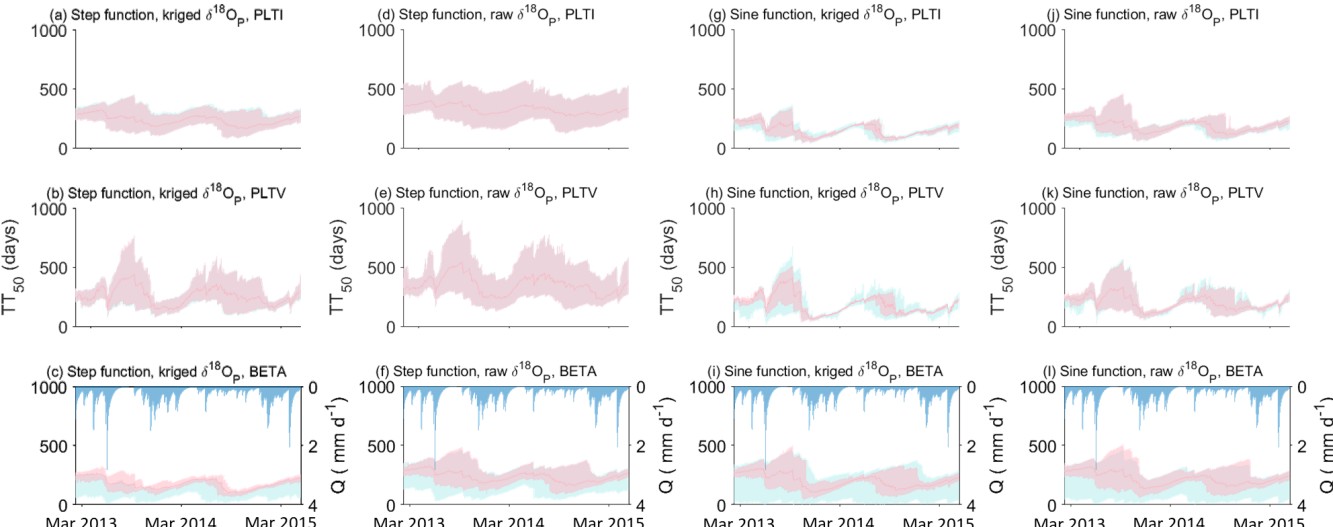

**Figure 5.** Predicted $TT_{50}$ for streamflow identified by 5% best KGE (blue) and by adding $F_{yw}^{est}$ (pink); the light blue (pink) line and the shaded area represent, respectively, the ensemble mean of all possible solutions and its variation range according to the 90% CI.

When considering the solutions identified by 5% best KGE, the catchment generally experienced a stronger affinity for young water (i.e., $k_Q$<1, or $\alpha$<1 and $\beta$>1), rather than old water (i.e., $k_Q$>1, or $\alpha$>1 and $\beta$<1). These findings are in agreement with other studies in the Upper Selke (Winter et al., 2020; Nguyen et al., 2021). Nonetheless, when PLTV is employed there is a considerable number of solutions indicating preference for both young and old water. Only a few solutions showed old water release, and this was 137% more prominent when using the sine interpolation and BETA, across all tested setups.

The use of $F_{yw}^{est}$ as additional constraint drastically changed the water release scheme for some of the tested setups. Figure 6 shows that the catchment mainly released old water when using the sine interpolation (Fig. 6g and h), especially with PLTV and BETA. Although $F_{yw}^{est}$ helped limit the uncertainty in the simulated $TT_{50}$ under specific model assumptions, $F_{yw}^{est}$ yielded contrasting results in the water release scheme, thus highlighting further sources of uncertainty related to those same setups.

## 5 Discussion

### 5.1 Uncertainty in TTD modelling

In this study we characterized the TTD uncertainty arising from model inputs (i.e., tracer data interpolated in time and space) and structure (i.e., SAS parameterizations). Our results show that the uncertainty (i.e., 90% CI) of the simulated $TT_{50}$ was firmly dependent on the choice for the model setup (Fig. 4), and we found that the 90% CI was primarily sensitive to the SAS parameterizations as well as temporal interpolation of $\delta^{18}O_P$, rather than spatial interpolation of $\delta^{18}O_P$. Uncertainty in $TT_{50}$ differed considerably between time-invariant (i.e. PLTI and BETA) and time-variant (i.e., PLTV) SAS functions. PLTI and BETA explicitly assume constant water selection preference over time, creating a moderately stable 90% CI with small



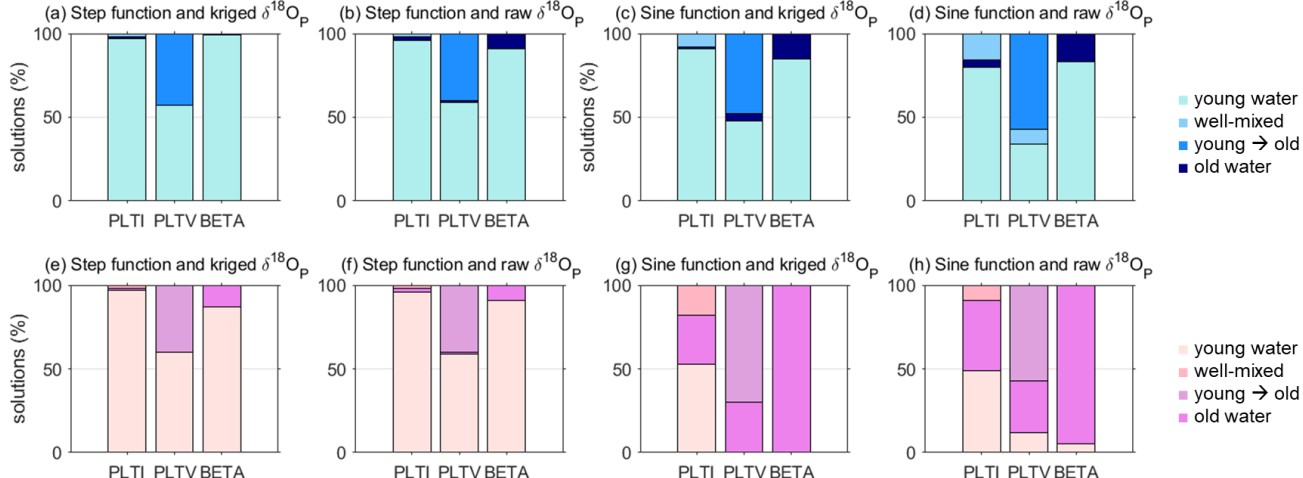

**Figure 6.** Percentage of behavioral solutions releasing water of different ages identified by 5% best KGE (blue) and by adding $F_{yw}^{est}$ (pink).

fluctuations. Moreover, for these SAS functions, the 90% CI is biased towards the long-term average discharge behavior, which might prevent a correct representation of the full $TT_{50}$ variability during precipitation events and droughts (Buzacott

et al., 2020). However, PLTI and BETA could be appropriate where the catchment release scheme is expected to be relatively constant.

On the other hand, including an explicit time dependence strongly affected the 90% CI with respect to constant assumptions. PLTV produced a wider 90% CI, notably during low flow conditions, which can hinder the ability of the TTDs to provide robust insights on flow and solute transport. This highlights the need to constrain PLTV with further data. In addition, the

exceptionally old flow components associated with a very large 90% CI might be a distortion of the actual $TT_{50}$, as they are known to be better estimated with reactive tracers rather than stable isotopes (Visser et al., 2019). Thus, PLTV-based $TT_{50}$ that are greater than the observed period (828 days) should be interpreted carefully.

Despite the sizeable 90% CI, PLTV could be more capable of capturing the hillslope activation and deactivation of flowpaths driven by the precipitation regime (Angermann et al., 2017; Loritz et al., 2017). Also, PLTV could infer information on the

time-variant water selection preference that could not be derived with PLTI and BETA. Indeed, PLTV showed the inverse storage effect, i.e., young water release during wet conditions rather than dry periods (Harman, 2015). This behavior may be the result of a rapid lateral transport due to rising water table (Pangle et al., 2017), and has been observed in many catchments (Benettin et al., 2017; Rodriguez et al., 2018; Wilusz et al., 2017, 2020).

Likewise, the high-frequency reconstruction of $\delta^{18}O_P$ estimates from monthly values via interpolation created further un-

certainty, as real data provide insights that are irreplaceable. The sine interpolation poorly reproduced heavily flashy rainfall events and only captured the average damped trend of the observed $\delta^{18}O_P$ samples (Fig. S2 in the Supplement). Hence, related results must be interpreted with caution as tracer data uncertainty may conceal more pronounced transport processes (Dunn et al., 2008; Birkel et al., 2010; Hrachowitz et al., 2011). Contrarily, the step function interpolation preserved the maxima in





the monthly observed $\delta^{18}O_P$ values, and reproduced their variation correctly. However, these results are based on this particular
isotope dataset; the sine interpolation may be suitably applicable in other circumstances.

The dissimilarities in the simulated $TT_{50}$ across the tested setups underline the importance of accounting for uncertainty in
model-based TTDs. The uncertainty analysis performed in this study was essential to best describe the parameter identifiability
and bounds of the behavioral solutions of each output variable. Furthermore, our results highlight the importance of gaining
a good quality for tracer datatasets and, possibly, employing the "true" model parameterization which correctly describes the
catchment area. The second point can be defined according to a precise conceptual knowledge of the catchment's functioning
(e.g., the geometry of the flow system) and information from previous studies in similar catchments.

### 5.2   On the value of young water fraction

The use of $F_{yw}^{est}$ succeeded in limiting the uncertainty in the predicted $TT_{50}$ obtained from the sine interpolation of $\delta^{18}O_P$
values and BETA. This reveals how these two setups combined are characterized by a high degree of uncertainty which
is propagated into the simulated $TT_{50}$. We acknowledge that this is due to the sine interpolation, which only captured the
average depleted trend of $\delta^{18}O_P$ values (Fig. S2 in the Supplement). In addition, BETA is characterized by numerous behavioral
solutions skewed towards short $TT_{50}$ values and, in turn, large $F_{yw}^{sim}$ (Fig. 4i and l), which might not correctly represent the
$TT_{50}$ values in the study area.

We recognize the robustness of $F_{yw}^{est}$ obtained by the sine regression to ensure reliable $TT_{50}$ values in this study. We
attribute this to the moderately small standard errors in $F_{yw}^{est}$ (0.07 and 0.08 for raw and kriged $\delta^{18}O_P$ values, respectively).
This is partly due to the fact that we used the same sampling period for $\delta^{18}O$ in precipitation and streamflow, which limits
the uncertainty of estimated $F_{yw}^{est}$. However, a higher temporal resolution of isotope samples rather than a coarser one could
prevent bias toward smaller $F_{yw}^{est}$ (Stockinger et al., 2016; Lutz et al., 2018). Also, to better represent the estimated $F_{yw}^{est}$ from
the sine regression, we could incorporate processes affecting the isotopic composition such as snowmelt (von Freyberg et al.,
2018; Ceperley et al., 2020). Despite these limitations, based on our analysis, we acknowledge the potential of using $F_{yw}^{est}$ for
improving TTD predictions and limiting their uncertainty.

### 5.3   TTD modelling: advantages and limitations

Our results provide visually plausible seasonal fluctuations of the predicted $\delta^{18}O_Q$ samples (Fig. 2), and satisfactory KGE
values (Fig. 3), despite the uncertainty arising from both model inputs and structure. This corroborates our findings for the
simulated $TT_{50}$ in the Upper Selke. The magnitude of the uncertainty resulting from different setups cannot be generalized,
but the overall approach for uncertainty assessment presented here could be extended to other areas and studies. However, we
recognize some limitations and below indicate possible reasons and, in turn, improvements that future work could achieve.

First, the limited length of the $\delta^{18}O$ timeseries might not describe the system accurately. Implementing longer timeseries
could improve the parameter identifiability and provide a closer approximation of the TTDs. Second, this study relied only
on stable water isotopes, which might underestimate the tails of the TTDs (Stewart et al., 2010; Seeger and Weiler, 2014).
Possible advancements could be reached by using decaying tracers varying over a larger timescale than stable water isotopes





(e.g., tritium, (Stewart et al., 2012; Morgenstern et al., 2015)), and imparting more information on old water. Next, future work should retrieve more information on *ET* and the initial storage $S_0$, whose parameters were poorly identified. However, this issue is common in transport studies that rely on measurements of instream stable water isotope (Benettin et al., 2017;

Buzacott et al., 2020). As a way forward, information on the *ET* isotopic compositions might help better constrain *ETET* parameters and assess their affinity for young/old water. Regarding constraining the range of $S_0$, further information can be gained knowing the geophysical properties of catchment (i.e., geological structure and physical characteristics) (Holbrook et al., 2014), and using decaying isotopes (Visser et al., 2019). Finally, future research should better explore the validity of $F_{yw}$ to limit the predictive uncertainty in TTDs, by exploring whether the effectiveness of $F_{yw}$ for TTD estimation depends on

catchment's attributes (e.g., size, annual precipitation, flow rates, soil and vegetation).

## 5.4 Implications of TTD uncertainties

This study characterized the uncertainty in TTDs, which summarize the catchment's hydrologic transport behavior, thereby comprise decisive information for water managers. The uncertainty in the predicted $TT_{50}$ has relevant implications for both water quantity and quality; the larger the 90% CI in the simulated $TT_{50}$, the greater the difference in the $TT_{50}$ values, which,

ultimately, implies distinct water release and solute export dynamics (McDonnel et al., 2010).

Uncertainty in TTDs may be crucial for characterizing the catchment's response to climatic changes (Wilusz et al., 2017). Considering the increasing severity of droughts (Dai, 2013), a catchment that largely releases young water might be more affected by droughts than a catchment whose stream is fed by relatively old water sources. A short $TT_{50}$ reveals a low drought resilience of the catchment, which could limit streamflow generation processes and change the instream water quality status.

Likewise, TTD uncertainty may affect the quantification of te modern groundwater age, i.e., groundwater younger than 50 years (Bethke and Johnson, 2008). According to (Jasecko, 2019), the correct identification of modern groundwater abundance and distribution can help determine its renewal (Le Gal La Salle et al., 2001; Huang et al., 2017), wells and depths most likely to contain contaminants (Visser et al., 2013; Opazo et al., 2016), and the part of the aquifer flushed more rapidly.

Uncertainty in TTDs also impacts on the fate of dissolved solutes, such as nitrates (Yang, X. et al., 2018; Nguyen et al.,

2021), pesticides (Holvoet et al., 2007; Lutz et al., 2017), and chlorides (Kirchner et al., 2000; Benettin et al., 2013). These solutes constitute a crucial source of diffuse water pollution in agricultural areas (Jiang et al., 2014; Kumar et al., 2020), as they are spread on the soil during the fertilization period. Exposure time of nitrates with the soil matrix has strong consequences for biogeochemical reactions, such as denitrification (Kolbe et al., 2019; Kumar et al., 2020). A short $TT_{50}$ suggests that water can be rapidly conveyed to the stream network (Kirchner et al., 2001), with limited time for denitrification. This explains the

elevated instream concentration and short-term impact of nitrate export compared to that of a longer $TT_{50}$, which is typically associated with old water release and low nitrate concentration (Nguyen et al., 2021). Similarly, pesticide transport is highly affected by the TTD uncertainty as a long $TT_{50}$ may decelerate pesticide degradation due to decreased microbial activity along deeper flowpaths (Rodríguez-Cruz et al., 2006). In other cases, a shorter $TT_{50}$ may limits the time for degradation causing a peak in the instream concentration (Leu et al., 2004). Overall, a longer $TT_{50}$ can delay or buffer the catchment's reactive solute

response at the outlet (Dupas et al., 2016; Van Meter et al., 2017). This creates a long-term effect of hydrological legacies and




a continuous problem with diffuse pollution of nitrates (Ehrhardt et al., 2019; Winter et al., 2020) and pesticides (Lutz et al., 2013), which can persist in the catchment for several years. Finally, TTD uncertainties also play an important role in chloride transport, although chlorides are commonly known to be conservative (Svensson et al., 2012). A short $TT_{50}$ may indicate rapid chloride mobilization, whereas a long $TT_{50}$ implies chloride persistence in groundwater; thereby chloride accumulates and is

released at lower rates, with impacts on the ecosystem functions, vegetation uptake and metabolism (Xu et al., 1999).

Understanding the uncertainty in TTDs is crucial for the aforementioned implications. In this study, we want to convey that it is essential to characterize the specific sources of uncertainty, which can stem from model inputs, structure and parameters, as well as their combined effects on the predicted TTDs. Uncertainty is omnipresent in TTD-based modeling, and we need to recognise it, especially when dealing with sparse tracer data and multiple choices for model parameterization, which affect the

calculated $TT_{50}$. We also acknowledge the potential of incorporating additional information on water ages based on $F_{yw}$, which has contributed to reduce $TT_{50}$ uncertainties.

## 6    Conclusions

This study explored the uncertainty in TTDs of streamflow, resulting from twelve model setups obtained from different SAS parameterizations (i.e., PLTI, PLTV and BETA), and reconstruction of the precipitation isotopic signature in time and space

via interpolation (step function vs. sine-fit, raw vs. kriged values).

We found satisfactory KGE values, whose differences across the tested setups were statistically significant, meaning that the choice of the setup matters and, as a consequence, distinct setups led to considerably different simulated $TT_{50}$ values. The choice between using time-variant or time-invariant SAS functions was crucial as the time-invariant SAS function generated a bias in the estimated $TT_{50}$ toward the long-term average discharge behavior, which might be appropriate for catchments

experiencing smooth changes in the hydrologic conditions. On the other hand, the time-variant SAS function captured the dynamics of the catchment wetness, resulting in very high seasonality of $TT_{50}$. However, the time-variant SAS function also produced a larger 90% CI, which indicates the need to constrain the function with additional data. Results from the temporal interpolation using a sine curve must be interpreted carefully as they poorly reproduced flashy events in precipitation, thus indicating that some more dynamic transport processes were not fully accounted for. Conversely, the step function interpolation

was able to intercept the measured $\delta^{18}O_P$ data. Dry conditions were another reason for uncertainty as indicated by the high variance in the simulated $TT_{50}$ values. Finally, the use of $F_{yw}$ as an additional constraint was promising in reducing $TT_{50}$ uncertainties, particularly when using the sine interpolation of $\delta^{18}O_P$ samples combined with BETA.

Our study provides new insights into the TTDs uncertainty when high-frequency tracer data are missing and the SAS framework is used. Regardless of the degree of efficiency or uncertainty, the decision on which setup is more plausible depends on a

full conceptual knowledge of the catchment functioning. We consider the presented approach as potentially applicable to other studies for enabling a better characterization of TTDs uncertainty, improving TTD simulations and, ultimately, informing water management. These aspects are particularly crucial in view of evermore extreme climatic conditions and increasing water pollution under global change.



*Code and data availability.* The model used in this study is presented at https://doi.org/10.5194/gmd-11-1627-2018. The iteratively re-
weighted least squares (IRLS) method used to get modelled daily kriged and raw isotope ($\delta^{18}$O) in precipitation with the sine interpolation,
as well as the estimated young water fraction ($F_{yw}$), is presented at https://doi.org/10.5194/hess-22-3841-2018. Hydrocliamtic timeseries,
$\delta^{18}$O data and interpolated $\delta^{18}$O timeseries can be accessed at https://zenodo.org/record/6630477#.YqL7wBpBxaQ.

*Author contributions.* AB conducted the model simulations, the analysis and interpretation of the results, and wrote the original draft of the
paper. SRL and RK designed and conceptualized the study, and provided data for model simulations. TVN provided technical support for
modeling and helped organize the structure and content of the paper. AB, SRL, RK and TVN conceived the methodology and experimental
design. All co-authors helped AB interpret the results. All authors contributed to the review, final writing and finalization of this work.

*Competing interests.* RK is a member of the editorial board of Hydrology and Earth System Sciences.



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
