# Peer review of "Uncertainty in water transit time estimation with StorAge Selection functions and tracer data interpolation"

_Hydrology and Earth System Sciences, 2022_

## Author Response (AR1)

We would like to thank the editor for allowing us to proceed with revising the manuscript, and we would like to express our appreciation to the reviewers for their valuable comments. We carefully considered all of the feedback we received and incorporated it into our revised manuscript, and have also made some minor changes of our own. Below you will find our point-by-point response (in blue text) to the reviewers' comments (in black text) as well as the changes we made in response to these comments (in bold blue text).

Referee #1

This paper presents an analysis of uncertainty in transit time distributions estimated using SAS functions, including that arising from the interpolation of input tracer data, and from the SAS function parameterization. Uncertainty of each configuration of model and input data is assessed from the range of predictions made by the top 5% of monte-carlo sampled parameter sets ranked by goodness-of-fit (KGE). The fraction of young water $F_{yw}$ obtained from the method proposed by Kirchner (2016) is used to further constrain the behavioral set. This paper aims to address an important gap in the literature. There is a need to better understand the uncertainty associated with SAS models, and how data can be best used to constrain them.

We thank the reviewer for acknowledging the important gap in the literature (uncertainty induced by SAS parameterization and input data) that we want to address in this manuscript.

However, I think there are two major problems with the approach used here, and I think the resulting conclusions are unsupported as a result.

- I don't think it makes sense to use $F_{yw}$ to constrain the SAS model parameterizations.

We appreciate this comment, and we understand the reviewer's concern that it seems it might not make more meaningful sense to use Fyw, derived using a relatively simple approach (the sine-wave fitting approach), to constrain the results already run with the best available data obtained from a more "elaborated" model (the StorAge Selection – SAS approach). This is especially true as we cannot know if Fyw from the sine-wave approach is better than that from the SAS approach or not. We note, however, that the main goal of our study is to highlight the uncertainty in SAS-based modeled results arising from model inputs, as well as underlying model structure and parameters – that have been not thoroughly evaluated yet in previous studies. The use of Fyw from the sine-wave fitting approach as an additional and minor part of the presented work, as an attempt to suggest a further metric that might be helpful in constraining the model simulations of an already calibrated SAS model.

The reviewer's comments have stressed the strong assumptions we have made in the use of Fyw as an additional model constraint. We agree that it may need a more elaborate procedure that considers the uncertainty in sine-wave fitted Fyw and corresponding age thresholds for young water (see below for further explanations) to relax some of these assumptions. Adding this to the revised manuscript would, however, put the focus too much on the use of Fyw and distract from the first and major part of the manuscript i.e. demonstrating the appreciable uncertainty in SAS

modeling. Hence, we have decided to discard the part about Fyw from this manuscript and instead plan to develop and illustrate this approach more thoroughly in a different study. model.

**We have excluded the section on Fyw from this manuscript.**

- I think the use of top 5% KGE to define behavioural parameter sets makes it impossible to meaningfully compare the uncertainty of each configuration

Thank you for this remark. As we understood, the reviewer suggests to use a fixed KGE value for defining the behavioral simulations, rather than fixing the sample size based on best 5% KGE, as we proposed.

Firstly, by doing this, we will run into the same problem raised by the reviewer – these behavioral simulations do not have the same range in goodness-of-fit i.e. KGE. In fact, if we define the behavioral simulations as those with KGE ≥ 0.5, the range of KGE with setup 1 is KGE = [0.5, 0.64], while for setup 4 IT is KGE = [0.5, 0.72] as it is possible to see in the range of behavioral KGE values in Fig. 2 of the original manuscript. Secondly, fixing the KGE threshold will lead to a different sample size per each model setup. For example, if we choose a fixed threshold limit of KGE ≥ 0.5, the behavioral solutions range between 1,300 and 2,700 across the 12 model setups. When looking at the uncertainty in the simulated outputs, the 90% confidence interval is wider for model setups that have a larger number of behavioral solutions than for those that have a smaller number. Therefore, a varying sample sizes would affect the uncertainty analysis. With a fixed sample size based on the 5% best KGE, we can ensure a meaningful comparison in uncertainty across the model scenarios. Also, we are still able to meet the requirement of a minimum acceptable KGE value (minimum KGE in the behavioral solutions across all tested setups is 0.57).

Despite this, we acknowledge that fixing the sample size is not necessarily better than imposing a threshold limit as there will be always a tradeoffs and pros/cons of each of the chosen approaches. However, given (i) the arguments provided above, (ii) the objective of our study (showing the uncertainty in the modeled outputs arising from model inputs, structure and parameters, not identifying the best simulations) and (iii) the large number of model setups explored i.e. 12, we find it more appropriate to use the top 5% simulations. Therefore, we would like to keep the definition of behavioral solution in the way we proposed in the revised manuscript. Nevertheless, during the revision, we will make it clear the reasons regarding the chosen criterion by providing the supporting motivation described above.

Also, as we understood from the reviewer, we cannot use the GLUE methodology if we consider the top 5% simulations as behavioral because "*each behavioral set would have a different total likelihood associated with it (if a formal likelihood were estimated)*". Therefore, in the revised manuscript, we will use the informal likelihood (the Sequential Uncertainty Fitting Procedure – SUFI-2; Abbaspour et al., 2004), an approach that has been widely used for estimating parameter uncertainty of eco-hydrological models (e.g., the Soil and Water Assessment Tools – SWAT, Arnold et al., 2012). In this way, we will estimate the uncertainty for the top 5% best parameters

which is described by a uniform distribution, and not by a formal likelihood such as done in GLUE. We have already checked differences in results with the SUFI-2 approach versus those with the GLUE approach, and we have found insubstantial differences in the model prediction uncertainty (see below for more details).

Reference:

Abbaspour, K. C., Johnson, C. A., & van Genuchten, M. T. (2004). Estimating uncertain flow and transport parameters using a sequential uncertainty fitting procedure. Vadose Zone Journal , 3 , 1340–1352. https://doi.org/10.2136/vzj2004.1340.

Arnold, J. G., Moriasi, D. N., Gassman, P. W., Abbaspour, K. C., White, M. J., Srinivasan, R., ... & Jha, M. K. (2012). SWAT: Model use, calibration, and validation. *Transactions of the ASABE*, *55*(4), 1491-1508.

**We have modified our approach for identifying the behavioral solution from the GLUE methodology to the SUFI-2 approach. A description of the new method can be found in lines 164-170:**

*The informal likelihood of the Sequential Uncertainty Fitting Procedure (SUFI-2, Abbaspour et al., 2004) was applied to account for uncertainty in the SAS parameter sets and resulting modeled estimates. In SUFI-2, the uncertainty in model parameters and simulated results is represented by a uniform distribution, which is gradually reduced until a specific criterion is reached. In our study, we calibrated the values of model parameters until the predicted output matched the measured tracer data to a satisfactory level, defined by an objective function. We employed as objective function the Kling-Gupta efficiency (KGE, Gupta et al., 2009), and once the criterion of KGE>=0.5 was satisfied, we defined a set of behavioral solutions for each model setup.*

**Furthermore, we have included an explanation for our decision to maintain the definition of behavioral solution based on the top 5% of simulations in terms of KGE. This can be found in lines 169-177:**

*However, since the aim of this study is to investigate the impact of various sources of uncertainty on simulated outputs, rather than to determine the best model setup based on the model efficiency, we decided to set a fixed sample size and narrow down those solutions generated by SUFI-2 in the previous step. Setting a fixed sample size ensures comparability of results across the twelve tested setups, as different sample sizes could influence the uncertainty analysis. For example, the greater the number of behavioral solutions, the wider the uncertainty band. At the same time, by fixing the sample size, we can still meet the requirement of a minimum acceptable KGE value (KGE≥0.5).*

*In this study, we determined the final behavioral solutions by using a fixed sample size that corresponds to the best 5% parameter sets and modeled results in terms of KGE. …*

**Major issues**

**Use of $F_{yw}$ to constrain SAS models**

- I do not think it makes sense to use the young water fraction obtained from the sine-wave ratio to constrain a SAS model. Kirchner's method for this is useful for obtaining rough estimates of the fraction of water that is roughly a quarter of a year old from tracer time series. The method might be robust (in some sense) but it isn't precise. SAS models are a more complex and sophisticated tool that have the *potential* to provide a much more precise estimate of water age distribution from the same data. It doesn't make sense to me to use the outputs of a rough-and-ready model to constrain the parameters of a more precise one.

Thank you for this comment. Please, refer to our response above for the proposed modifications in the revised manuscript.

Here, we want to add that additional complexity to constrain models does not necessarily lead to a better result than the use of simple models. This, for example, has been demonstrated and supported in the hydrological community through different studies (Michaud and Sorooshian, 1994, Orth et al., 2015, Merz et al., 2022). Also, the reviewer argues that the sine-wave fitting approach is not precise. Although we cannot generally falsify this statement, it is also difficult to prove that it is fully correct given that the level of "preciseness" is difficult to assess for both approaches (sine-wave fitting and SAS functions). To our knowledge, there are no studies proving that Fyw from the sine-wave fitting approach is not "precise". Conversely, the sine-wave fitting approach has increasingly been acknowledged in the past years for estimating Fyw (Jasecko et al., 2016, Lutz et al., 2018; von Freyberg et al., 2018 , Stockinger et al., 2019; Gallart et al., 2020). However, we agree that there is a need for a more rigorous testing to better understand, which approach provides a better estimate of Fyw based on the available data (same as done for the transit times in a recent paper by Benettin et al., 2022). Since this topic is out of the scope of current work, we will revise our work - excluding the part on Fyw discussion - and focus on the uncertainty in the SAS models.

References:

- Michaud, J. and Sorooshian, S. (1994) Comparison of Simple versus Complex Distributed Runoff Models in a Midsized Semi-Arid Basin. Water Resources Research, 30, 593–605, https://doi.org/10.1029/93WR03218.

- Orth, R. Staudinger, S.I. Seneviratne, J. Seibert, & M. Zappa (2015) Does model performance improve with complexity? A case study with three hydrological models. J. Hydrol., 523, 147–159, https://doi.org/10.1016/j.jhydrol.2015.01.044.

Merz, R., Miniussi, A., Basso, S., Petersen, K. J. & Tarasova, L. (2022) More Complex is Not Necessarily Better in Large-Scale Hydrological Modeling: A Model Complexity Experiment across the Contiguous United States. BAMS, E1947–E1967, https://doi.org/10.1175/BAMS-D-21-0284.1.

Jasechko, S., Kirchner, J. W., Welker, J. M., & McDonnell, J. J. (2016). Substantial proportion of global streamflow less than three months old. Nat Geosci, 9, 126–129, https://doi.org/10.1038/ngeo2636.

Lutz, S. R., Krieg, R., Müller, C., Zink, M., Knöller, K., Samaniego, L., & Merz, R. (2018). Spatial patterns of water age: using young water fractions to improve the characterization of transit times in contrasting catchments. Water Resour. Res., 54 , 4767–4784, https://doi.org/10.1029/2017WR022216.

von Freyberg, J., Allen, S. T., Seeger, S., Weiler, M., & Kirchner, J. W. (2018). Sensitivity of young water fractions to hydro-climatic forcing and landscapeproperties across 22 Swiss catchments. Hydrol. Earth Syst. Sci., 22 , 3841–3861, https://doi.org/10.5194/hess-22-3841-2018.

Stockinger, M. P., Bogena, H. R., Lücke, A., Stumpp, C., & Vereecken, H. (2019). Time variability and uncertainty in the young water fraction in a small headwater catchment. Hydrol. Earth Syst. Sci., 23, 4333–4347, https://doi.org/10.5194/hess-23-4333-2019. 2018.

Gallart, F., Valiente, M., Llorens, P., Cayuela, C., Sprenger, M., & Latron, J. (2020). Investigating young water fractions in a small mediterranean mountain catchment: Both precipitation forcing and sampling frequency matter. Hydrol. Process., 34 , 3618–3634, https://doi.org/10.1002/hyp.13806.

Benettin, P., Rodriguez, N. B., Sprenger, M., Kim, M., Klaus, J., Harman, C. J., et al. (2022). Transit time estimation in catchments: Recent developments and future directions. Water Resour. Res., 58, e2022WR033096, https://doi.org/10.1029/2022WR033096.

**We have excluded the section on Fyw from this manuscript.**

- I believe the fact that the authors do find that $F_{yw}$ has power to constrain the SAS parameters is largely because the uncertainty in the associated age threshold $\tau_{yw}$ is not accounted for. The method that $F_{yw}$ relies on is based on a variety of assumptions, including that the inputs are sinusoidal and that the transit time distribution is approximately a gamma distribution. Two important *and distinct* sources of uncertainty here are:

- The threshold age of the young water fraction $\tau_{yw}$ is not 75 days, as suggested by the authors. Rather it depends on the shape parameter of the assumed gamma distribution. As Figure 10 of Kirchner (2016) shows, for a shape parameter of 0.2 it is around 40 days, while for a shape parameter of 2 it is more like 100 days. This considerable uncertainty is not accounted for in the present paper.

- The estimates of amplitudes $A_q$ and $A_p$ obtained from fitting sinusoids to the observed tracer timeseries are uncertain, and that uncertainty ought to be estimated and propagated into uncertainty in $F_{yw}$. The authors may have accounted for this (if I understand the brief statement on line 165) but they claim that in doing so they have also accounted for the uncertainty in $\tau_{yw}$, which is not the case. These errors are independent of each other. The errors

obtained for $F_{yw}$ were only 0.07-0.08 (line 325), which I suspect contributes far less uncertainty than the 60-day window bracketing $\tau_{yw}$ paper.

Thank you for the above observations on the uncertainty in Fyw and the young water threshold ($\tau_{yw}$). Although we will remove the Fyw part from the manuscript, we acknowledge that the uncertainty in $\tau_{yw}$ was not properly addressed in our original manuscript. We agree with the reviewer that not only the uncertainty in Fyw (which we accounted for in the original manuscript) should be considered, but also in $\tau_{yw}$ (which we did not) when fitting the sine function to the tracer data in inflow and outflow.

**We have excluded the section on Fyw from this manuscript.**

- Furthermore, the theory behind $F_{yw}$ and $\tau_{yw}$ rests on the assumption that flows through the system are steady, the transit time distribution is invariant, and that the input signal is a perfect sinusoid. These are not the case in general in real watersheds, which results in additional epistemic uncertainty into the estimates of $F_{yw}$ and $\tau_{yw}$. These particular sources of uncertainty do not necessarily apply to the SAS models, since they can allow for variable flows, variable transit time distributions, and make use of the observed input signal.

Thank you for this comment. Although we will remove the Fyw part from the revised manuscript, we would like to comment on the aspect of the steady state assumption - correctly highlighted by the reviewer. In the original manuscript we first estimated the transient daily transit time distribution (TTD) and then derived the marginal TTD, from which we calculated the Fyw values. By estimating the marginal TTD, we assume to reflect the steady state behavior, though admittedly not perfect, but this could be a reasonable approach. To the aspects of the input signal and the transit time distribution, we agree with the reviewer that the isotope signal in inflow and outflow does not perfectly follow the sinusoidal as the marginal TTD might not perfectly follow a gamma distribution. Therefore, we acknowledge that the approach presented in the original manuscript has some limitations and there is uncertainty in Fyw (which we accounted for in the original manuscript) and in $\tau_{yw}$ (which we did not) when fitting the sine function to the tracer data in inflow and outflow.

However, it could also be argued the other way around: SAS functions have uncertainties (e.g. lack of agreement on which model parameterization to use, equifinality of parameters, assumptions regarding age distributions of evapotranspiration) that, in contrast, do not apply to Fyw obtained with the sine-wave fitting approach. Indeed, we explored and highlighted some of these uncertainties in the current study (i.e. tracer data interpolation and choice for SAS parameterization), which have not been emphasized in detail in previous studies.

**We have excluded the section on Fyw from this manuscript.**

- In fact, it is possible to reproduce the model used to justify Kirchner's method as a SAS model. This can be done by approximating the flows as constant, replacing the inputs concentrations with sinusoids, and choosing a SAS function whose corresponding steady-state TTD is a gamma. From

this perspective $F_{yw}$ and $\tau_{yw}$ can be viewed as outputs of a particular SAS model parameterization run with degraded data. Why should the results of that parameterization be used to constrain other parameterizations run with the best available data?

Thanks for this observation. We agree with the reviewer that, being the SAS parameters already calibrated and being the model already run with the best available data, there may be no reason for further constraining the model with any additional metrics e.g. Fyw. For this reason, as we have already augmented in the first response, we will remove the Fyw part from the manuscript.

**We have excluded the section on Fyw from this manuscript.**

**Use of top 5% KGE as the 'behavioural' parameter set**

- The use of the top 5% KGE as the 'behavioural' parameter set makes it impossible to make meaningful comparisons between the different parameterizations (i.e. PLTI, PLTV, BETA). This is because the range of goodness-of-fit (i.e. the KGE) of each model's behavioral set depends on the size of the pool from which it was taken, in addition to how well it actually fits the data. The range of KGE in the top 5% depends on the assumed prior distribution of the parameter set, since that determines what the 5% is a percentage of. Since each parameterization has fundamentally incommensurate parameters, there isn't an obvious way to normalize for this dependence across different parameter spaces. As a result each behavioral set would have a different total likelihood associated with it (if a formal likelihood were estimated). Comparing these different behavioral parameterizations therefore makes no sense, since they have been held to different standards.

- One consequence of effectively holding each parameterization to a different standard is that the error associated with the more flexible parameterizations (PLTV, BETA) is larger than that associated with the less flexible one (PLTI), when we would expect the opposite to hold. This is particularly true given that PLTI represents a special case of both PLTV and BETA (when $k_{Q1}=k_{Q2}=k$ and when $\alpha=k, \beta=1$ respectively). However, as seen in Figure 2 the behavioral sets of BETA (and to a lesser extent PLTV) seem to include models that are considerably worse fits to the data than the worst models in the behavioral set of PLTI.

- To make meaningful comparisons between different parameterizations the analysis would need to be redone with a standard for 'behavioral' that is consistent across the different parameterizations. This might be as simple as choosing a cutoff value of KGE to define the behavioral set, but it would likely change the resulting conclusions about the merits of each parameterization.

Thank you for this observation. Please, refer to our response above for the reasons why we want to keep the definition of behavioral solution based on the 5% best simulations in terms of KGE, and the proposed modifications in the revised manuscript.

Here, we just want to show that there are no significant differences in the results with the SUFI-2 approach compared to those with the GLUE approach for quantifying the model prediction

uncertainty (e.g. as shown below for the simulated instream isotope and median transit times for one of the 12 tested model setups).

[Figure]

We have modified our approach for identifying the behavioral solution from the GLUE methodology to the SUFI-2 approach. A description of the new method can be found in lines 164-170:

*The informal likelihood of the Sequential Uncertainty Fitting Procedure (SUFI-2, Abbaspour et al., 2004) was applied to account for uncertainty in the SAS parameter sets and resulting modeled estimates. In SUFI-2, the uncertainty in model parameters and simulated results is represented by a uniform distribution, which is gradually reduced until a specific criterion is reached. In our study, we calibrated the values of model parameters until the predicted output matched the measured tracer data to a satisfactory level, defined by an objective function. We employed as objective function the Kling-Gupta efficiency (KGE, Gupta et al., 2009), and once the criterion of KGE>=0.5 was satisfied, we defined a set of behavioral solutions for each model setup.*

Furthermore, we have included an explanation for our decision to maintain the definition of behavioral solution based on the top 5% of simulations in terms of KGE. This can be found in lines 169-177:

*However, since the aim of this study is to investigate the impact of various sources of uncertainty on simulated outputs, rather than to determine the best model setup based on the model efficiency, we decided to set a fixed sample size and narrow down those solutions generated by SUFI-2 in the previous step. Setting a fixed sample size ensures comparability of results across the twelve tested setups, as different sample sizes could influence the uncertainty analysis. For*

*example, the greater the number of behavioral solutions, the wider the uncertainty band. At the same time, by fixing the sample size, we can still meet the requirement of a minimum acceptable KGE value (KGE≥0.5).*

*In this study, we determined the final behavioral solutions by using a fixed sample size that corresponds to the best 5% parameter sets and modeled results in terms of KGE. …*

**Minor issues**

- Line 57: The gamma distribution has also seen some use

We will add the gamma distribution to the list of commonly used parameterizations employed to approximate the SAS functions.

**We have added the gamma distribution and have written:** *Finally, SAS functions, employed to model TTDs, must be parameterized and commonly used parameterizations are the power law (Benettin et al., 2017; Asadollahi et al.,60 2020), beta (van der Velde et al., 2012; Drever and Hrachowitz, 2017) and gamma (Harman, 2015; Wilusz et al., 2017) distribution.* **This can be found at lines 65-66.**

- Line 64: I don't think that the statement that $F_{yw}$ is useful for short-term data is quite right, since the method does require data covering multiple cycles of sinusoidal variation to fit to reliably

As we will remove the Fyw part in the revised manuscript, this phrase will not be part of the revised manuscript.

**We have excluded the section on Fyw from this manuscript.**

- Line 111: $S_{T_0}$ is a function of age: $S_{T_0}(T)$

We will change $S_{T_0}$ to $S_{T_0}(T)$.

**We have written $S_{T_0}(T)$. This can be found at lines 111 and 114.**

- Line 130: $k_{Q1}$ and $k_{Q2}$

Here we do not use the subscript *Q* referring to streamflow, because we describe the parameters of SAS functions in general, without referring to a specific flux. Therefore, we prefer to leave *k* rather than *kQ* in lines 127-131. However, in the rest of the text, we specify which parameterization (e.g. PLTI and PLTV) we apply to each flux (i.e. streamflow and evapotranspiration), so we write *kQ*, *kQ1*, *kQ2* and *kET*.

**We have kept the *k* parameter rather than with the subscript *Q* or *ET*. This can be found at lines 129-131.**

- Table 2: Why are $k_{Q1}$ and $\alpha$ grouped together? Same with $k_{Q2}$ and $\beta$

There is no specific reason: we simply decided to group *kQ1* with *alpha* and *kQ2* with *beta* because the two correspond to the shape parameters of PLTV and BETA, respectively. Upon the reviewer's suggestion, we will disaggregate them into separate rows (in Table 1) in the revised manuscript.

**We have disaggregated $k_{Q1}$ and $\alpha$, as well as $k_{Q2}$ and $\beta$ from the same rows in Table 2.**

- Line 188: Is $TT_{50}$ is the median of the *backward* transit time distribution $p_Q(T,t)$ as defined in equation (5)? In that case this statement is incorrect, and should be "the maximum time elapsed *since* the youngest 50% of the water in outflow first entered the catchment", or perhaps "the age that half the outflow is older than, and half younger than, as measured from the time it fell as precipitation".

Here we consider the backward formulation of the transit time distribution. We will clarify this in line 188 and modify the definition of median transit time accordingly.

**We have re written: …*and backward median transit time (TT50 (days), i.e., the maximum time elapsed until the youngest 50% of the infiltrated water is transferred to the outflow)*. This can be found at lines 180-181.**

- Figure 3: A legend explaining the colors and a reference to Table 1 would aid interpretation here

We will add a legend and a reference to Table 1 in Fig. 3.

**We have added a legend and a reference to Table 1 and Fig. 3.**

- Line 221: Parameters for the *SAS function* of $Q$...

We will add "SAS functions" in the revised manuscript.

**We have re written: *Parameters for the SAS functions of Q (i.e., kQ, kQ1, kQ2, α and β)*… This can be found at line 213.**

Referee #2

In this study, the authors studied the uncertainty in transit time estimation. Two sources of uncertainty were considered: the assumed shape of the StorAge Selection (SAS function (and the uncertainty in the associated parameters) and the interpolation scheme for the precipitation tracer data. The reported uncertainty is large, resulting in a 90% confidence interval between 286 – 895 days for the median transit time. The uncertainty was greater in dry conditions than in wet conditions. The uncertainty depended more on the SAS function parameterization and the temporal interpolation of the precipitation tracer data than the spatial interpolation of the precipitation tracer data. Importantly, the authors argued that it could be useful to utilize the young water fraction, Fyw, in estimating the SAS function parameters, as it could constrain the SAS function and reduce uncertainty.

Thank you for this summary. The main intent of this paper is to explore the uncertainty in the and simulated isotopes and median transit times arising from the differences in model inputs, structure, and parameters. In our case, we want to study the uncertainty when different interpolation techniques are used to construct the high-frequency behavior of tracer data, for the application of SAS-based modeling framework. The use of Fyw is a subordinate objective as an attempt to suggest a further metric that might be helpful in constraining the model simulations of an already calibrated SAS model.

Though understanding the uncertainty in the SAS function and transit time is important, it is unclear what readers could learn from this manuscript in its present form other than the summarized results above for the specific catchment. As the results were not discussed enough in detail, it is not easy to think about their implications (see major comment 1). The suggestion of using the young water fraction in the SAS function estimation is interesting, but the authors' argument regarding using it must be more convincing (see major comment 2). In addition, some additional potential sources of uncertainty should be considered or mentioned explicitly (see my major comment 3). Recent advances in estimating transit should be mentioned (see my major comment 4). Thus, I think a significant revision is required before this manuscript can be considered for publication in HESS again.

Thank you for raising these issues. We agree with the reviewer that these issues were not addressed properly in the original manuscript; therefore, we will revise the manuscript accordingly. Please see below our detailed responses to these comments.

- Discussion of the results

I think it is necessary to discuss the results further to make the implications of this study clearer. The current manuscript focuses more on describing the results for the specific catchment and dataset than discussing the results, so it is not easy to think about those implications. For example, why is the uncertainty in the estimated TTD (or the median transit time) large? Why is the uncertainty greater under drier conditions? Why does the spatial interpolation method not substantially affect the water age simulation? Without discussing that type of question for each finding, it is not easy to truly understand the described results.

Thanks for this comment. We would like to point out that the original manuscript has already an entire subsection (5.4) that discusses the implications of uncertainty in water transit times for water quantity and quality studies. However, we agree with the reviewer that further discussion, including the points mentioned above, should be included in the revised manuscript, and they are outlined in the following.

Previous studies used only a specific SAS function and/or specific tracer data interpolation technique. Our work shows that there could be a wide range of different results (in terms of water age and instream isotopes, as well as model performances), and parameter uncertainties due to distinct model setups regarding SAS parameterization and tracer data interpolation technique, at least for our study area. With this, we want to encourage similar studies in other catchments to

explore these uncertainties and examine whether or not they can influence their conclusions/implications for water quality and quantity management.

We would like to specifically address the questions posed in the comment in the revised manuscript as follows:

1) Firstly, the uncertainty analysis is done among all 12 tested setups corresponding to different combinations of spatial/temporal data interpolation techniques and SAS functions. We found that the uncertainty in the median transit time is large (Fig. 4), which is mainly due to the temporal interpolation of isotopes, which resulted in very different reconstructed input data depending on whether the step function or sine interpolation is used (Fig. S2). This explains why the simulated water transit times are different between the two interpolations or, in other words, why the uncertainty in the water transit time is large. Then, the choice of which SAS function to use also leads to differences in the simulated water transit times (Fig. 4), thus large uncertainty associated with the model parameterization. For example, choosing a time-invariant function created a time series with more moderate fluctuations (Fig. 4a, c, d, f, g, i, j and l), while the choice of a time-variant function led to more marked fluctuations (Fig. 4b, e, h, k), specifically between wet and dry conditions.

Secondly, the uncertainty analysis is done among the behavioral solutions within each single model setup. Here, we also found a large uncertainty which might be due to the poor identifiability of some model parameters (Table S1), such as the evapotranspiration parameter and the storage parameter - the latter being a key factor that deserves further attention in the application of the SAS framework for modeling outflow isotope signals.

2) We found greater uncertainty under drier conditions especially visible when using time-variant SAS functions (Fig. 4b, e, h, k), because the uncertainty increases along with the median transit time (Fig. 4), which is actually longer during drier periods. When the catchment is wetter nearly all flow paths are active and contribute to the streamflow. Thus, flows are generally "smoother" and water ages are easier to constrain. Conversely, under dry conditions only selected flow paths are active, usually those longer flow paths carrying older water to the stream partly through a drier soil zone, where the flow is more erratic as the conductivity is controlled by the soil moisture. Hence, the flows in the soil matrix are less uniform, which could make it more difficult to constrain these older water ages. We will further elaborate this part in the revised manuscript.

3) The spatial interpolation method did not substantially affect the simulations (at least in our particular case) because there is no big difference between kriging and raw isotopes as it is possible to see in Figs. S1 and S2.

Reference:

Benettin, P., Soulsby, C., Birkel, C., Tetzlaff, D., Botter, G., and Rinaldo, A. (2017), Using SAS functions and high-resolution isotope data to unravel travel time distributions in headwater catchments, Water Resour. Res., 53, 1864– 1878, doi:10.1002/2016WR020117.

**We have better discussed the implications of our work, and stated how there could be a range of possible outcomes in terms of water ages, instream isotopes, and model performance, as well as parameter uncertainties resulting from different model setups. We have encouraged similar studies in other catchments to investigate these uncertainties and their impact on water quality and quantity management recommendations. This can be found at lines 369-375:**

*While previous studies have used only a specific SAS function and/or specific data fitting technique, here we show that there could be a wide range of different results in terms of water ages, model performances and parameter uncertainty. This is due to the specific choice regarding SAS parameterization and tracer data interpolation. With this, we want to convey that uncertainty is omnipresent in TTD-based models, and we need to recognise it, especially when dealing with sparse tracer data and multiple choices for model parameterization. Therefore, we want to encourage future studies to explore these uncertainties in other catchments and different geophysical settings, with the final aim to investigate whether these uncertainties may affect the conclusions of water quantity and quality studies for management purposes.*

**We also have thoroughly explained how we conducted the uncertainty analysis across the 12 different tested setups. We have emphasized which aspects or conditions have a greater (e.g., choice in SAS function, temporal interpolation and dry conditions) or smaller (e.g., spatial interpolation) impact on the uncertainty of median transit times. This can be found at lines 261-310.**

Overall, it needs to be clarified if the use of Fyw constrained the SAS function parameters in the right way.

Thank you for this comment. We agree with the reviewer that, in general, the robustness in Fyw from the sine-wave fitting approach to constrain SAS function parameter should be checked, for example, by looking at the simulated isotopes and model efficiencies from the SAS framework after constraining with Fyw. This can allow us to know if Fyw reduces the model outputs towards the 'right' or 'wrong' values. However, developing this in a more elaborate approach, which also considers the uncertainties associated with Fyw and the young water threshold ($\tau_{yw}$) would go beyond the scope of this paper. So, we do not intend to present the use of Fyw in the revised manuscript, but we plan to develop a more elaborate way to account for uncertainties in Fyw and $\tau_{yw}$ in another study. Please also refer to the more elaborate response in the other reviewer's comments document.

**We have excluded the section on Fyw from this manuscript.**

The authors somehow decided to state that the estimated young water fraction indicates the fraction of water younger than $\tau_{yw} = 75$ days (in L171). However, as the authors mentioned, for example in L171 and L256, the method of Kirchner (2016) does not provide a single value for $\tau_{yw}$ that can be utilized universally. Rather, it varies with the shape of TTD. While the authors argued that the

arbitrary decision is okay since they considered the uncertainty in the estimated Fyw (in L 171-172), that argument was made without clear reasonings that support it.

Thank you for this observation. We acknowledge that the uncertainty in τyw was not properly addressed in the original manuscript. We agree with the reviewer that, in general, there is the need to consider the uncertainty in both Fyw (which we accounted for in the original manuscript) and τyw (which we did not) when fitting the sine function to the tracer data in inflow and outflow. However, as we have already said, there is need to develop this in a more elaborate way that would go beyond the scope of this paper, so we do not intend to present this in the revised manuscript. Please also refer to the more elaborate response in the other reviewer's comments document.

**We have excluded the section on Fyw from this manuscript.**

Also, it needs to be clarified if the estimated Fyw based on the method of Kirchner is a good estimate that can be used to constrain the SAS function. The method of Kirchner is based on a set of assumptions. It seems like the authors want to argue that it's okay to use the estimate regardless of all assumptions because the estimated uncertainty of Fyw is low, but I think it would be better if the authors could provide more concrete arguments to convince readers why it's okay. Why is the uncertainty low? And how does the low uncertainty support that the estimate is a good estimate regardless of all the assumptions? In addition, the method approximates the precipitation and outflow tracer signal using sinusoidal functions, which was shown to be not a good approximation for the precipitation tracer data by the authors in this manuscript (e.g., in L305-308).

Thank you for this comment. As explained above, in the revised manuscript we do no longer present this approach. We agree with the reviewer that, in general, the assumptions of the sine-wave fitting approach do not apply to SAS modelling. For example, the isotope signal in inflow and outflow might not perfectly follow the sinusoidal and the marginal transit time distribution might not perfectly follow a gamma distribution. Therefore, there is uncertainty in the use of Fyw from the sine-wave fitting approach. We partly accounted for it in the original manuscript, but there is need of a more elaborate method that would go beyond the scope of this paper. Please also refer to the more elaborated response in the comments document of the other reviewer.

**We have excluded the section on Fyw from this manuscript.**

It also seems that the uncertainty in Fyw could depend on the temporal resolution of data (if the finer resolution data shows more deviation from the sinusoidal signal) and other properties that the authors mentioned in L348-350. Overall, based on the limitation discussed by the authors (in L348-350), I feel that the authors are unsure whether the utilization of Fyw will be useful for other datasets.

Thank you for raising this point. It is true that collecting temporally refined tracer data could potentially help infer a more robust Fyw from the sine-wave fitting approach. However, it has been demonstrated that Fyw is the only approach that can robustly estimate some age statistics for discontinuous and low-frequency tracer time series for at least 2-3 years (Benettin et al., 2022). Low-frequency measurements are more readily available than high-frequency measurements,

especially over a wide spatial domain. For this reason, Fyw has the advantage to be much more largely applicable, and its use from less temporally refined tracer data should be acknowledged and further explored. There is certainly a need to account for its uncertainties but, as we will no longer include Fyw, there is no need to make this clearer in the revised manuscript.

Reference:

Benettin, P., Rodriguez, N. B., Sprenger, M., Kim, M., Klaus, J., Harman, C. J., et al. (2022). Transit time estimation in catchments: Recent developments and future directions. Water Resour. Res., 58, e2022WR033096, https://doi.org/10.1029/2022WR033096.

**We have excluded the section on Fyw from this manuscript.**

- A minor comment related to Fyw

L163-164: The method used to estimate Fyw was described too briefly. For example, L163-165 is not enough for readers to understand the method.

Thank you for this observation. As we will remove the Fyw part, so there is no need to make this point clearer in the revised manuscript.

**We have excluded the section on Fyw from this manuscript.**

- Other sources of uncertainty

I believe that the uncertainty in precipitation, discharge, and evapotranspiration rates could propagate into the uncertainty in the estimated SAS function. The list of potential sources of uncertainty provided by the authors (L39-43) needs to include them. It would be helpful for readers if the authors provided a more concrete list of potential sources of uncertainty. Also, it would be essential to provide why this manuscript, where the authors consider only a few sources of uncertainty, is still useful.

Thank you for mentioning this. We agree with the reviewer that the uncertainty in precipitation, discharge, and evapotranspiration rates could propagate into the uncertainty in the estimated SAS functions. However, in the current study, we do not consider these sources of uncertainty as we use the hydrologic simulations from the mesoscale Hydrologic Model (mHM, Samaniego et al. (2010); Kumar et al. (2013); Zink et al. (2017)), which is an established model to simulate daily discharge and evapotranspiration time series. However, we agree with the reviewer's suggestion to include in the revised manuscript this type of uncertainty in the list of potential sources of uncertainty affecting transit time-based models.

We have decided to focus on only a few sources of uncertainty that we think are the most significant and critical for SAS modelling, and also the most linked directly to the questions of data interpolation and SAS parameterization. Firstly, since there is no general agreement on which SAS function to use, we explored the uncertainty generated by the use of different SAS functions (i.e. model parameterization and parameters). Secondly, as SAS modelling requires continuous time series of input tracer data, we tested how different gap-filling techniques (i.e. temporal

interpolations) affect the model results. Finally, as the SAS models depend on the type of input data, we tested what happens when using regionalized or non-regionalized isotopic datasets (i.e. spatial interpolations). In the revised manuscript, we will make this part clearer in the Introduction and Discussion by emphasizing the reasons why we decided to explore these specific sources of uncertainty.

Reference:

Samaniego, L., Kumar, R., and Attinger, S. (2010) Multiscale parameter regionalization of a grid-based hydrologic model at the mesoscale, Water Resour. Res., 46, W05 523, 10.1029/2008WR007327, 2010.

Kumar, R., Samaniego, L., and Attinger, S. (2013) Implications of distributed hydrologic model parameterization on water fluxes at multiple scales and locations, Water Resour. Res., 49, 360–379, 10.1029/2012WR012195.

Zink, M., Kumar, R., Cuntz, M., and Samaniego, L. (2017) A high-resolution dataset of water fluxes and states for Germany accounting for parametric uncertainty, Hydrol. Earth Syst. Sci., 21, 1769–1790, 10.5194/hess-21-1769-2017.

**We have included precipitation, discharge, and evapotranspiration rates as potential sources of uncertainty that can impact the results of transit time-based models. This can be found in lines 43-44:**

*Additionally, uncertainty in the driving hydroclimatic fluxes such as precipitation, discharge, and evapotranspiration could propagate into the uncertainty of the modelling results.*

**We have clarified the reasons behind our decision to focus on a limited number of sources of uncertainty in SAS modeling. This can be found in lines 55-71:**

*… there are other aspects particularly significant for SAS modelling causing uncertainty in the simulated TTDs, which have not yet been thoroughly investigated. First, isotope data are generally sparse globally in space and time (von Freyberg et al., 2022), due to laborious and costly sampling campaigns limited to well-equipped areas (Tetzlaff et al., 2018). As SAS models require continuous time series of input tracer data, different methods for temporal interpolation could be used to fill gaps in isotope values in precipitation; consequently, the interpolated input data is subject to uncertainty. Furthermore, the input data of SAS models is influenced by whether the tracer data in precipitation are collected at a single location within the catchment, or at multiple locations. In the latter scenario, there is a need to account for the spatial variability of tracer composition in precipitation, which is commonly done via spatial interpolation. Choosing data from one approach (i.e., tracer data from a single location) over the other (i.e., multiple tracer data spatially interpolated) can potentially result in different resulting TTDs. Finally, SAS functions, employed to model TTDs, must be parameterized and their functional forms need to be specified a-priori. Commonly used forms are the power law (Benettin et al., 2017; Asadollahi et al., 2020), beta (van der Velde et al., 2012; Drever and Hrachowitz, 2017)*

*and gamma (Harman, 2015; Wilusz et al., 2017) distributions. However, there is no general agreement on which SAS function should be used since the hydrological processes that control the patterns and dynamics of the subsurface vary across catchments. Therefore, the most convenient approach is to simply rely on a specific parameterization over another, and estimate its parameters (Harman, 2015). All of these aspects, related to model input, structure and parameter, induce uncertainty in the simulated TTDs. To date, the role of these individual uncertainty sources and their combined effect on the modeled TTDs have not been adequately discussed.*

**Also, at lines 259-260:**

*In this study, we characterized the TTD uncertainty arising from some significant and critical aspects for the SAS modelling. These aspects are also the most directly linked to data interpolation and SAS parameterization that we explored in this work.*

- Missing new methods of estimating TTDs

There have been some recent advances in the estimation of TTDs that are not discussed in this manuscript. The method of Kirchner (2019) and the method of Kim and Troch (2020) can estimate time-variable (or state-dependent) TTDs without assuming their form a priori. The estimated TTDs can be converted to the SAS functions. Thus, some descriptions of the motivation of this study, such as what is in L55-59, need to be revised. useful.

Thank you for noting this. In the revised manuscript we will integrate the motivation of our study with the suggested literature.

**The suggested literature on recent advances in estimating TTDs has been added to the introduction at lines 38-39:**

*Recent research has introduced new models for representing time-variant TTDs, for example allowing for the estimation of TTDs without making prior assumptions about their shape (Kirchner, 2019; Kim and Troch, 2022), ...*

- Minor comments

fSAS/rSAS: I think the authors should make it clear that they are discussing only the fSAS (fractional SAS) function. Another form of the SAS function, the rank SAS (rSAS) function, may have different uncertainty characteristics, especially because of the difference in how the storage is considered

In our study SAS functions are expressed as function of the normalized age-ranked storage (i.e., fractional SAS functions). We will clarify this in both Methods and Discussions.

**We have clearly explained that we used the fractional SAS functions in our study. This can be found at lines 123-124 and 278-282, respectively:**

*Here, they are expressed as probability density functions in terms of the normalized age-ranked storage $P_S(T,t)$ (-), also known as fractional SAS functions (fSAS): …*

*However, in this study we discussed the fractional (fSAS) functions, while another form of the SAS functions, such as the rank SAS (rSAS) functions, may have different uncertainty characteristics. This is mainly due to the difference in how the storage is considered, because fSAS functions are expressed as function of the normalized age-ranked storage, which is equal to the cumulative residence time, while rSAS functions depend on the age-ranked storage, which is the volume of water in storage ranked from youngest to oldest (Harman, 2015).*

Naming of the "BETA" case: Better to name the case more clearly. While the beta distribution is used without any state dependency or time-variability in this study, several studies utilized state-dependent beta distribution that can consider the time-variable flow pathways (e.g., Van der Velde et al., 2015). Thus, it could confuse readers when the authors state something like "BETA could be appropriate where the catchment release scheme is expected to be relatively constant" (in L290-291). In this manuscript, the time variability is mentioned explicitly in the case names only for the power law cases (PLTI and PLTV).

We will say explicitly that we tested this study for a time-invariant beta distribution, and will re name the BETA parameterization as BETATI (i.e. time-invariant beta).

**In our study, we have renamed the BETA parameterization as BETATI to explicitly state that it has been utilized without any time-variability, in order to avoid any potential confusion and accurately reflects our methodology.**

L11-12: Make it clear that this confidence interval is for the median transit time.

We will clarify this.

**We have specified that the number of days presented as uncertainty for the median transit times refers to the confidence interval. This can be found in lines 10-11.**

*We found a large uncertainty in the simulated TTDs, represented by a large range of variability in the 95% confidence interval of the median transit time varying between …*

L113: V(t) (mm) "is"

We will correct it.

**We have adjusted the verb to be singular or plural as needed to match the subject. This can be found at line 113:**

*…. V(t) (mm) are the storage variations …*

L185: When using GLUE, the authors determined the behavioral parameters using an arbitrary criterion. The top 5% of the parameters, in terms of KGE, were selected as the behavioral parameters. However, there is no statement to support the decision regarding the criterion. It may be better for readers if the authors could explain why the criterion was chosen and whether this seemingly arbitrary choice is okay.

Thank you for pointing this out. We acknowledge that we have not supported the reasons why we chose to define a behavioral solution based on the top 5% of the parameters in terms of KGE. As

also said in the response to the other reviewer, in this work we explore the impact of different model setups by checking the uncertainties in the simulated outputs. The goal is not to identify the best model efficiency or setup. Given this objective, and considering the large number of model setups used, we find it appropriate to define a behavioral solution based on a fixed sample size identified with the lower bound dependent on the maximum KGE for each model setup, in our case 5%. In the revised manuscript we will make it clear the reasons regarding the chosen criterion by providing the supporting motivation described above. However, in the revised manuscript we will apply the SUFI-2 approach rather than the GLUE approach, as in the former the uncertainty of parameters is described by a uniform distribution and not by a formal likelihood (see answer to other reviewer for further explanations).

**We have modified our approach for identifying the behavioral solution from the GLUE methodology to the SUFI-2 approach. A description of the new method can be found in lines 164-170:**

*The informal likelihood of the Sequential Uncertainty Fitting Procedure (SUFI-2, Abbaspour et al., 2004) was applied to account for uncertainty in the SAS parameter sets and resulting modeled estimates. In SUFI-2, the uncertainty in model parameters and simulated results is represented by a uniform distribution, which is gradually reduced until a specific criterion is reached. In our study, we calibrated the values of model parameters until the predicted output matched the measured tracer data to a satisfactory level, defined by an objective function. We employed as objective function the Kling-Gupta efficiency (KGE, Gupta et al., 2009), and once the criterion of KGE>=0.5 was satisfied, we defined a set of behavioral solutions for each model setup.*

**Furthermore, we have included an explanation for our decision to maintain the definition of behavioral solution based on the top 5% of simulations in terms of KGE. This can be found in lines 169-177:**

*However, since the aim of this study is to investigate the impact of various sources of uncertainty on simulated outputs, rather than to determine the best model setup based on the model efficiency, we decided to set a fixed sample size and narrow down those solutions generated by SUFI-2 in the previous step. Setting a fixed sample size ensures comparability of results across the twelve tested setups, as different sample sizes could influence the uncertainty analysis. For example, the greater the number of behavioral solutions, the wider the uncertainty band. At the same time, by fixing the sample size, we can still meet the requirement of a minimum acceptable KGE value (KGE≥0.5).*

*In this study, we determined the final behavioral solutions by using a fixed sample size that corresponds to the best 5% parameter sets and modeled results in terms of KGE. …*

L198: "more positive": I am not sure if "more positive" is the right expression here. The isotope ratios are always negative in the data.

We will replace ''more positive'' with ''less negative''.

**We have updated the phrasing in our manuscript to use "less negative" instead of "more positive" at lines 188-189 when referring to isotope ratios:**

*The results reveal how the predicted $\delta^{18}O_Q$ values enveloped the measured isotopic signature by reproducing its seasonal fluctuations, with depleted (i.e., more negative) values in winter and enriched (i.e., less negative) values in summer.*

L188-190 I believe that the authors in general talked about the "backward" transit time distribution throughout this manuscript. However, the sentence in L188-190 is not clearly written if they talked about "forward" TTDs or the "backward" TTDs. Please revise.

Here, we consider the backward formulation of the transit time distribution. We will make it clear in the revised manuscript.

**We have re written: *…and backward median transit time (TT50 (days), i.e., the maximum time elapsed until the youngest 50% of the infiltrated water is transferred to the outflow).* This can be found at lines 180-181.**

Figure 2: I think it would be better to enumerate the subfigures using numbers (based on Table 1) instead of alphabets.

As we understand it, the HESS guidelines for language and typesetting for submission state that subfigures labels should be enclosed in parentheses around lowercase letters (e.g. (a), (b), etc.). Therefore, we would like to keep the enumeration of subfigures using letters. To be consistent between subfigures and model setups we will enumerate the setups displayed in Table 1 using letters instead of numbers. Figure 3 will be changed accordingly.

**We have re-enumerated the different configurations using letters instead of numbers to maintain consistency with the setup settings between Table 1 and Fig. 3.**

L290-291: Not clear if this trivial sentence is necessary.

We would like to keep this sentence as it is relevant to this study in order to inform when specific SAS parameterization should be used. For clarity, we mean that the catchment preference to discharge water of a certain age is time-invariant with time-invariant SAS functions (PLTI and BETA) as these functions do not consider time variability as a function of the catchment wetness (such as PLTV does). Therefore, their use might be more appropriate for catchments experiencing relatively constant hydrologic conditions without highly pronounced seasonality.

L298-303: This paragraph is also trivial and not necessary for the manuscript.

We will delete this paragraph.

**We have deleted the paragraph about the inverse storage effect.**

L305: I think the figure illustrating the interpolation results is more important for this manuscript than some unnecessary paragraphs mentioned above.

We will put more focus on this part by addressing the temporal interpolation of stable water isotopes in precipitation.

**We have expanded our discussion on the impact of tracer data interpolation techniques on the uncertainty of simulated median transit times. This can be found in lines 283-298:**

*Likewise, the high-frequency reconstruction of δ18OP estimates from monthly values via interpolation created further uncertainty that would not arise when using real high-frequency data. The sine interpolation poorly reproduced flashy rainfall events and only captured the average damped trend of the observed $\delta^{18}O_P$ samples (Fig. S2 in the Supplement). Hence, related results must be interpreted with caution as tracer data uncertainty may conceal a more pronounced hydrological response (Dunn et al., 2008; Birkel et al., 2010; Hrachowitz et al., 2011). Contrarily, the step function interpolation preserved the maxima in the monthly observed $\delta^{18}O_P$ values, and reproduced their variation correctly. Nonetheless, the results obtained in this study are based on this particular isotope dataset, while the sine interpolation may be better applicable in other circumstances. Overall, the temporal interpolation of tracers resulted in largely differing reconstructed input data depending on whether the step function or sine interpolation were used (Fig. S2 in the Supplement). This explains why the simulated $TT_{50}$ is different between the two interpolations or, in other words, why the uncertainty in $TT_{50}$ is large. On the contrary, the spatial interpolation method did not strongly affect the simulated $TT_{50}$ as the trend in the time series was similar when using kriged (Fig. 4a-c and g-i) or raw (Fig. 4d-f and j-l) $\delta^{18}O_P$. This could be attributed to minor differences between kriged and raw isotopes (Figs. S1 and S2 in the Supplement). Nonetheless, there was a larger 95% CI of $TT_{50}$ when using raw rather than kriged $\delta^{18}O_P$, and this was particularly visible when the step function interpolation was used (Fig. 4a-f). Therefore, the spatial interpolation of δ18O in precipitation from different locations resulted in an apparent reduction of uncertainty in $TT_{50}$.*

L345: Typo: "ETET"

We will change this to "ET".

**We have corrected this at line 332.**

L360: Typo: "te modern"

We will change this to "the".

**We have corrected this at line 345.**

L395: "smooth changes" are unclear.

We will replace it with ''little seasonality in the hydrological conditions''.

**We have replaced ''smooth changes'' with ''little seasonality in the hydrological conditions'' in referring to the use of a time-invariant SAS function. This can be found at lines 383-384:**

*These functions may be more appropriate for those catchments experiencing relatively little seasonality in the hydrological conditions.*

---

## Author Response (AR2)

Editor # Comment & Responses

Dear authors,

based on the comments received for your revised manuscript, it turns out that a few additional elements of information would be required. These relate essentially to methodological aspects - mainly related to the choice of a sinusoidal function in the SAS model (i.e., still justified or not after the removal of the YWF from the study), and the choices made for rainfall tracer data (one site vs. multiple sites). I consider these elements requiring a minor to moderate revision. I am looking forward to receive the revised version of your contribution.

Best regards,

Laurent Pfister

We thank the editor for approving the revision of our manuscript. In the revised version, we have explained the reasons for using sine interpolation for stable water isotopes in precipitation, while acknowledging its limitations. We have also tested the GAM approach as suggested by the reviewer and presented our findings; our major conclusions regarding the uncertainty of SAS-based modeling remain intact, regardless of the interpolation method used. We have provided a comprehensive explanation for using both raw (data from one station only) and kriged isotopes in precipitation and clarified the selection of raw isotopes at the catchment outlet. Finally, we have addressed the impact of different spatial representations of isotopes in precipitation on SAS-based results more effectively.

Please find below our point-to-point responses (text in blue) to the reviewer's comments (text in black) and implemented modifications (text in italic blue) in the original manuscript. The line number in this document refers to the track-changes document.

Reviewer #2 Comment & Responses

This revised manuscript shows improvements over the previous one. The authors have addressed some (but not all) of the issues in the previous manuscript by removing the use of the young water fraction and have instead focused on investigating the uncertainties associated with the SAS function modeling. This study reiterates a common concern about the uncertainty of SAS models when trace data sets are limited.

The authors considered the uncertainties arising from temporal interpolation and spatial interpolation of the rainfall tracer data, as well as the uncertainty arising from the predetermined formula describing the SAS function. Although investigating the model uncertainty arising from the temporal interpolation has been previously addressed, as acknowledged by the authors (e.g., Buzacott et al., 2020), it seems that the incorporation of spatial rainfall tracer data (in addition to the consideration of the predetermined shape of the SAS function) could offer a novel contribution.

The authors concluded that the temporal interpolation method affects the result significantly (L511-512), while the spatial interpolation method did not substantially affect the uncertainty (L516-519). (Note that the line number in this document is based on the track-change document.)

However, there are several study designs that I find unconvincing and thus I recommend a major revision. Please refer to my comments below. There may be something I'm missing, and if so, I think it would be relatively easy for the authors to respond to my comments.

We thank the reviewer for taking the time to review our manuscript, providing constructive comments and for raising relevant issues. We acknowledge them and have carefully addressed them in the revised manuscript. Please find our detailed responses to each comment below.

1. Temporal interpolation: On the use of the sinusoidal function

The purpose behind using the sinusoidal function in the SAS function model remains unclear. While I comprehend that the authors aimed to demonstrate the model's sensitivity to the choice of temporal interpolation method, it seems apparent that the sinusoidal function could not capture the observed input tracer signal well. Consequently, it is unclear why this particular method, which misses many features in data, was used with the SAS function model. Once this question arose, I found it challenging to follow the manuscript smoothly. In the previous manuscript, I speculated that, by using the sinusoidal function, the authors want to develop an argument related to the young water fraction (which utilizes the sinusoidal function), but this issue becomes apparent after the removal of the young water fraction from the manuscript.

Although the sinusoidal function has been utilized in TTD modeling, e.g., when estimating the young water fraction, its application there is to focus on capturing only dominant features like seasonality. Note that, for the estimation of young water fraction, the outflow tracer time series is also approximated using the sinusoidal function. I am unsure if capturing only seasonality is a valuable practice in the SAS function modeling. Also, I am not sure about the meaning of uncertainty when the input tracer data that only approximates the seasonality is used to model the outflow tracer data that contain more detailed features.

Thank you for bringing up these concerns on the use of the sine function interpolation.

We acknowledge the uncertainty in sine interpolation as it misses detailed temporal features of the tracer dataset, such as individual observed peak values, but rather captures basic characteristics of the temporal pattern, such as seasonality. However, the dominant trend in long-term $\delta^{18}Op$ is often the seasonal trend (Feng at al., 2009), which can be effectively captured using a sine-wave function (Kirchner, 2016).

In our study, we compared two relatively simple, rather opposing temporal interpolation approaches, one emphasizing seasonality (sine-wave function) and one individual measurements (step function). This distinction was highlighted in lines 171-173:

*By employing step function and sine interpolation as techniques to reconstruct tracer data in precipitation, we aim to analyze the effects on SAS-based results from two relatively simple, rather opposing approaches: one focusing on individual measurements and the other on seasonality.*

Our findings show that both step function and sine interpolation yielded satisfactory goodness-of-fit (Fig. 5 in the revised manuscript) and effectively captured the trend in simulated instream $\delta^{18}O$ (Fig. 4 in the revised manuscript). This highlights the appropriateness of capturing the dominant seasonal trend in instream $\delta^{18}O$. However, using sine interpolation comes with limitations as individual observations are generally overestimated (Fig. 4g-l in the revised manuscript; consequently, it is important to acknowledge these uncertainties. Nonetheless, our results indicate that interpolation methods that precisely capture all observed data (e.g., step function) do not necessarily yield better SAS-based results as a whole. In fact, combining step function with raw $\delta^{18}Op$ resulted in larger uncertainty in simulated $TT_{50}$ (Fig. 6d-f in the revised manuscript). This reflects the purpose of our study which is to showcase two relatively simple, opposing choices for temporal interpolation to highlight that both (and thus potentially also many other methods) give acceptable model results. Hence, we emphasize the need for a comprehensive exploration of the uncertainty range, rather than relying solely on a specific model setup which may be subjective.

It should also be noted that uncertainty associated with sine interpolation found in this study is specific to the isotopic dataset used. Under different circumstances, where the isotopic dataset has a more pronounced sinusoidal trend (for example, see Fig. 1 of Von Freyberg et al., 2018) and/or higher temporal resolution, where the sinusoidal pattern should be more evident, sine interpolation may be more suitable and yield better results. However, investigating these aspects goes beyond the scope of the study.

Overall, in the revised manuscript we have acknowledged the limitations of sine interpolation raised by the reviewer, and expressed them in lines 321-329:

*The sine interpolation effectively captured the dominant features of the observed $\delta^{18}Op$, such as seasonality. Consequently, sine interpolation successfully reproduced the seasonal trend in instream $\delta^{18}O$, although simulations overestimated the measurements (Fig. 4g-l). On the other hand, sine interpolation poorly reproduced rainfall isotopes during short-term flashy events and missed detailed characteristics of the tracer dataset by smoothing the variability in the observed $\delta^{18}Op$ (Fig. 3). As a result, high values of tracer data in precipitation are underestimated, whereas low values are overestimated. It is critical to recognize these limitations when interpreting modeling results as uncertainty in the simulated $\delta^{18}Op$ may conceal a more pronounced hydrological response of the system (Dunn et al., 2008, Birkel et al., 2010, Hrachowitz et al., 2011).*

Moreover, we acknowledge that sine-wave fitting of seasonal isotopic cycles is commonly used for estimating the young water fraction. However, the sine-wave function has been used in other studies to describe temporal variation in $\delta^{18}Op$ (McGuire & McDonnell, 2006; Allen et al., 2019) due to the sinusoidal pattern characterizing $\delta^{18}O$. We have clarified this point in lines 161-165:

*... Second, we used a sine interpolation due to the fact that $\delta^{18}Op$ samples typically exhibit pronounced seasonal variations with more depleted values in winter than in summer (Fig. 2). The sine-wave function has been used in several studies to describe temporal variation in isotope in precipitation (McGuire & McDonnell, 2006; Feng et al., 2009; Allen et al., 2019).*

Reference:

Allen, S. T., Jasechko, S., Berghuijs, W. R., Welker, J. M., Goldsmith, G. R., and Kirchner, J. W.: Global sinusoidal seasonality in precipitation isotopes, Hydrol. Earth Syst. Sci., 23, 3423–3436, https://doi.org/10.5194/hess-23-3423-2019, 2019.

Feng, X., Faiia, A. M., and Posmentier, E. S.: Seasonality of isotopes in precipitation: A global perspective, J. Geophys. Res, 114, D08 116, https://doi.org/10.1029/2008JD011279, 2009.

Kirchner, J. W.: Getting the right answers for the right reasons: Linking measurements, analyses, and models to advance the science of hydrology, Water Resour. Res., 42, W03S04, https://doi.org/10.1029/2005WR004362, 2006.

von Freyberg, J., Allen, S. T., Seeger, S., Weiler, M., and Kirchner, J. W.: Sensitivity of young water fractions to hydro-climatic forcing and655 landscape properties across 22 Swiss catchments, Hydrol. Earth Syst. Sci., 22, 3841–3861, https://doi.org/10.5194/hess-22-3841-2018, 2018.

Buzacott et al. (2020) employed a sophisticated temporal interpolation method in the SAS function modeling, namely the Generalised Additive Model (GAM), to perform gap-filling and estimate the uncertainty of the gap-filled data. They subsequently explored how this estimated input uncertainty propagated through the SAS model. I believe that this approach provided more informative insights

compared to utilizing multiple methods that include the uncommon practice of fitting the sinusoidal function in the SAS function modeling.

Thank you for raising this interesting point. We tested the GAM for the reconstruction of both kriged and raw δ¹⁸Op used in this study; our findings can be seen below.

[Figure]

Fig 1: Predicted δ¹⁸Op via GAM with kriged (left) and raw (right) data.

[Figure]

Fig. 2: Simulated instream δ¹⁸O from GAM.

[Figure]

Fig. 3: Simulated TT₅₀ from GAM.

Our results show that in our analysis GAM generally produced a closer fit to the input tracer data (Fig. 1 in this document) compared to sine interpolation (Fig. 3 in the revised manuscript). However, when analyzing the temporal evolution of simulated instream δ¹⁸Op (Fig. 2 in this document) and TT₅₀ (Fig. 3 in this document) using the same SAS function and spatial representation of δ¹⁸Op, GAM did not lead to

significantly different SAS model results in comparison to sine interpolation (Fig. 4g-l and 6g-l in the revised manuscript). Furthermore, the magnitudes of uncertainty in $\delta^{18}Op$ and $TT_{50}$ are generally comparable, except in the case of $TT_{50}$ when PLTV is utilized.

We conclude that the improved input reconstruction by GAM does not provide significantly improved SAS-model output (probably due to the conceptual simplifications inherent to the SAS-model) and, in turn, new insights or conclusions in our study. Indeed, similar to what found when using step function and sine interpolation, the time-variant SAS functions (PLTI and BETATI) show moderate fluctuations in the $TT_{50}$ time series compared to the time-variant function (PLTV), whereas the uncertainty is generally higher during low flow conditions. Therefore, in this study we decided to maintain step function and sine interpolation as the two techniques for reconstructing tracer data in time, as they allowed us to explore the specific effects of general seasonality (sine function) and individual measurements (step function), while evaluating their influence on generating distinct results. However, we acknowledge the existence of other interpolation methods, such as the GAM suggested by the reviewer, and have included the results for GAM in the Supplement of the revised manuscript for comparison (lines 339-346):

*It is important to note that alternative methods, such as Generalized Additive Models (GAM; Buzacott et al., 2020) offer other options for interpolating tracer data. We conducted further tests with the SAS model using GAM to reconstruct both kriged and raw $\delta^{18}Op$ using smoothing functions; this provides a more sophisticated approach than the intuitive methods used in this study. However, the results, available in the Supplement, show that while GAM provided more detailed reconstructed input tracer data (Fig. S1), it did not significantly alter the SAS-based results (Figs. S2 and S3) or yield any new insights or conclusions with respect to using step function and sine interpolation. Therefore, we conclude that while highly resolved input data may seem appealing, it does not lead to substantial benefits for the SAS-based output, supposedly due to the conceptual simplifications in the SAS model.*

If the authors still intend to present the results using the sinusoidal function, it is essential for them to provide a compelling argument justifying the necessity of using the sinusoidal function over other methods that have been applied for the gap-filling, despite the concerns and points I have raised earlier. Without that, I worry that others could argue that the presented large uncertainty (or the significant differences in the median transit time, e.g., L511-512) is just because the temporal interpolation method utilizing the sinusoidal function was poorly performing.

Please see our answers above for further clarification. Here we would like to add that because of the uncertainties and potential errors in the observed data, determining the best temporal interpolation method is a challenge and is outside the scope of this study; our primary objective is to explore the uncertainties arising from different, commonly used choices in the model setup. Additionally, our main results regarding the uncertainty of the SAS modeling approach remain consistent, even when comparing our more simplistic reconstruction methods of $\delta^{18}Op$ with the use of GAM. Finally, despite the limitations, sine interpolation reasonably captures the essential characteristics of the tracer input signal for the SAS model at a hand.

2. Spatial Interpolation: Conclusion regarding the use of spatial rainfall tracer data

Despite having rainfall data from multiple locations, the authors have chosen to only present results for two cases: 1) the SAS model result using the data collected around the outlet, and 2) the result obtained by using spatially interpolating values based on data collected at 24 locations using kriging. The decision to focus solely on these two rainfall tracer time series is unfortunate and appears to underutilize the full potential of the dataset.

Thank you for raising these relevant concerns on the spatial distribution of isotopes in precipitation.

In our study, we investigated two contrasting spatial representations of $\delta^{18}O$ to compare their effects on model performance, results, and uncertainty. We examined a simple approach using single point $\delta^{18}O$ measurements taken at the catchment outlet and a more sophisticated method involving spatial interpolation of $\delta^{18}O$ with kriging based on multiple locations, including stations outside the catchment boundary to capture regional precipitation patterns. This analysis allowed us to evaluate the influence of spatial variability on SAS-based results. Exploring these two contrasting approaches in spatial representation of $\delta^{18}O$ aligns with the use of two contrasting temporal interpolation methods, one focusing on seasonality and the other on individual measurements. We have clarified this point in lines 150-152:

*By considering these two options for spatial representation of $\delta^{18}Op$, we intend to assess the range of uncertainty in the simulated outputs between two opposing cases i.e., raw isotopes representing the simplest approach and kriged isotopes derived from a more sophisticated method.*

While there are certainly other choices for tracer data or interpolation techniques that could be explored, we had to make a choice for our experimental design and selected these two cases to provide insights into the research question, i.e., are SAS models affected by whether $\delta^{18}Op$ is collected at a single location within the catchment or at multiple locations? We have emphasized this point in lines 152-155:

*While there are other possibilities for spatial representation of $\delta^{18}Op$, our choice allows us to effectively address our research question regarding the effects on SAS models of tracer data in precipitation collected at a single location within the catchment or spatially interpolated from multiple locations.*

It remains unclear whether the authors would reach the same conclusion when utilizing other rainfall time series collected at different locations (for their 'raw' case). Consequently, it is unclear what meaningful insights can be gleaned from the presented results.

What was the reasoning behind exploring the two cases (e.g., for the 'raw' case, why is the location close to the catchment outlet selected)?

Figure 4 in this document shows raw $\delta^{18}Op$ measured at various locations in the Upper Selke, revealing minimal spatial variability. In our study, we particularly focused on using raw data from the outlet. While we could have opted for another location, we chose the station close to the gauge at the outlet in the lowlands, assuming that at this location a precipitation collector would most likely be found in most catchments. Logistically, sampling instream $\delta^{18}O$ at the outlet is common practice as it is the location where all precipitation inputs across the catchment are integrated into streamflow. For convenience, also precipitation is often monitored at or near the gauging station at the outlet. We acknowledge that this approach may not be the best practice in catchments where several precipitation stations exist, as it has its own limitations, and we stated this at lines 353-356. However, it is important to note that our goal was not to determine the best approach for the spatial representation of $\delta^{18}Op$ (even the absence of interpolation) but rather we aimed to compare two contrasting methods to examine differences in SAS-based outcomes and uncertainties.

We have provided further clarification on this point in lines 139-142:

*The selection of $\delta^{18}Op$ at the outlet assumes a precipitation collector close to the stream gauge at the outlet, which is a common occurrence in many catchments for logistical reasons. Indeed, the outlet, where instream $\delta^{18}O$ is sampled, serves as location where all precipitation inputs across the catchment are integrated. For convenience, precipitation monitoring is also often conducted at or near the gauging station at the outlet.*

[Figure]

Fig. 4: Measured δ¹⁸Op (left) from the four precipitation collectors in the Upper Selke (right).

Why do the two rainfall tracer time series presented in Figure S1 are similar?

In our study, we found that differences between the δ¹⁸Op time series reconstructed using step function and sine interpolation methods were similar, except for a slightly more depleted signal in the kriged δ¹⁸Op due to the inclusion of isotopes from higher-altitude locations within the kriging process. This could be different for other catchments and it is outside the scope of the study to test the accuracy/representativeness of specific interpolation methods.

While there were no significant differences in the evolution of the TT₅₀ time series and instream δ¹⁸O between the two methods for spatial representation of δ¹⁸Op, higher uncertainty was observed when using raw δ¹⁸Op. This highlights the potential advantages of spatial interpolation over the simplistic use of δ¹⁸Op from a single location, particularly with step function. This finding shows how relying solely on model performance (Fig. 5 in the revised manuscript) may not reveal the increased uncertainty associated with the single-station method (or other chosen methods). By incorporating uncertainty analysis, it is possible to make informed decisions about the most suitable representation/interpolation method for a specific application.

Taking this into consideration, we have revised the text to emphasize the distinct implications of using raw and kriged δ¹⁸Op in SAS models.

Lines 12-14:

*The large 95% CI and the notable differences across the tested setups highlight the sensitivity and, in turn, uncertainty of predicted TT₅₀ associated with the model parameterization, choice of temporal interpolation of input data, hydrologic conditions and non-spatially interpolated δ¹⁸Op.*

Lines 347-358:

*The spatial representation of δ¹⁸Op values had limited impact on the overall pattern of simulated TT₅₀ as the TT₅₀ time series were comparable with both kriged (Fig. 6a-c and g-i) or raw (Fig. 6d-f and j-l) δ¹⁸Op. Nonetheless, the spatial interpolation of δ¹⁸Op from different locations resulted in a reduction in the uncertainty of TT₅₀, which was particularly evident with step function. This difference may be attributed to the fact that the Upper Selke is a large (mesoscale) catchment with a substantial gradient in elevation, and, as a consequence, a single point measurement for δ¹⁸Op may be generally overly simplistic.*

*This finding highlights the importance of considering not only the model performance in terms of goodness-of-fit (Fig. 5; raw values with a step function interpolation produced higher KGE values), but also the uncertainty range in predicted TT₅₀.*

Lines 376-379:

*Furthermore, our results highlight the importance of gaining tracer datasets of good quality (i.e., tracer data with a finer temporal resolution), considering the spatial variability of the isotopic composition in precipitation and, possibly, employing a model parameterization that best describes the catchment-specific storage and release dynamics.*

When should we expect a spatially interpolated value to be similar to at-a-point measurement and when we shouldn't?

The spatially interpolated $\delta^{18}O$ used in this study were obtained by applying kriging with altitude as external drift (work done in Lutz et al., 2018) on a set of stations that also includes some stations outside the catchment boundaries, after which a catchment-average of the kriged values was obtained. This was done because the isotopic composition can vary with altitude due to factors such as temperature. The kriging process predicts isotopes at unknown locations by using isotopes which are known only at given locations of the study area. In our study, we found that the values for kriged and spatially averaged $\delta^{18}Op$ are slightly more negative than raw $\delta^{18}Op$, as the raw $\delta^{18}Op$ at the multiple locations (partially in the mountainous area) considered in the kriging process are more negative than the raw $\delta^{18}Op$ measured at the catchment outlet (in the lowland part) of the study area.

The above points have been incorporated at lines 144-149:

*The spatial interpolation was conducted in Lutz et al., (2018) using $\delta^{18}Op$ from 24 precipitation collectors spread over the larger Bode region, and altitude as external drift. In a further step, the kriged $\delta^{18}Op$ were weighted with spatially distributed monthly precipitation to obtain representative estimates for the study catchment. In our study, the kriged (and spatially averaged) $\delta^{18}Op$ resulted in slightly more negative values than the raw $\delta^{18}Op$ from the catchment outlet (Fig. 2 and 3) because of the inclusion of more depleted $\delta^{18}Op$ values from locations with higher altitudes during the kriging process.*

I personally like the arguments provided in L392-395 as they read like the additional information (the information used in the kriging) is valuable in the SAS function model (though unclear if the authors would get to the same conclusion if they chose another location for the 'raw' case). However, it is not clearly stated in the Conclusion section (e.g., in L516-519). That part of the Conclusion may be to be modified.

We revised the part about spatial interpolation in the Conclusions section on lines 4566-470:

*Finally, there was a comparable pattern in the modeled results when using kriged versus raw isotopes, but the kriged values yielded an uncertainty reduction in $TT_{50}$. This highlights the potential advantage of spatially interpolated values over a single measurement representative of the entire catchment, particularly in mesoscale catchments varying in elevation.*

3. Other comments

Figures S1 and S2: The figures illustrate one of the most important results, i.e., the interpolation results. It would greatly enhance the manuscript's comprehensibility if these figures were included in the main manuscript not in the supplement, as they are essential to understand the study.

Figures S1 and S2 have been moved from the Supplement to the main text and now represent Figures 2 and 3, respectively.

L224: The meaning of '2.5% and 97.5% CIs' is unclear.

We have clarified the meaning of '2.5% and 97.5% CIs' by writing in lines 202-206: *To assess the range of possible behavioral solutions and understand the level of uncertainty associated with the solutions, we computed the 95% Confidence Interval (CI), which was derived by calculating the values of the 2.5% and 97.5% percentile of the cumulative distribution in the time series of the output variables. These values represent the lower and upper limits of the CI, respectively.*

L227: The definition provided for 'backward' median transit time seems to align more with the definition of 'forward' median transit time.

The median transit time is the time it takes for half of the water particles to leave the system; the backward representation relates to the ages of water particles leaving the system at a given time, thus they are considered in terms of the distribution of entrance times. Therefore, at lines 208-209 we have revised the text on the backward median transit time as: *the time it takes for half of the water particles to leave the system as streamflow at the catchment outlet.*

L356-357: I would recommend removing such trivial and somewhat unrelated results and interpretations from the manuscript.

L505-507: The same argument repeated in the Conclusions section. It is unclear if this trivial statement is relevant to the uncertainty explored in this study.

L356-357, L505-507: I noticed that I have already provided the same comment for the previous manuscript. I still do not see the relevance of this argument in this study. If the authors think that the argument is necessary, please explain how you arrived at the argument based on the findings presented in this study.

In the revised manuscript we have removed results and interpretations referring to the suitability of a time-invariant or time-variant SAS function depending on the presence or absence of pronounced seasonality in hydrological conditions.

Figure 3: Please correct the legend.

We have corrected the legend.

L402: The term 'uniform' may not be appropriate here. It seems that the authors are referring to potential event-to-event variations in the flow pathway during low-flow conditions. (Maybe I am wrong here.)

The term 'uniform' in the phrase '...flows in the soil matrix are less uniform...' refers to the variability of flow pattern and direction under dry conditions. We have clarified this in lines 362-378:

*… Conversely, under dry conditions, when usually only longer flowpaths carrying older water are active (Soulsby and Tetzlaff, 2008; Jasechko et al., 2017), water partially flows through a drier soil zone where flow is more erratic (i.e. flow directions and patterns can vary widely) as the conductivity is controlled by soil moisture. As a result, wet areas can be patchy and water flows preferentially at certain locations only, as opposed to spatially uniform flow through the soil matrix; this might make it more challenging to constrain older water ages.*

'raw' vs. 'kriged': Just a suggestion, it may be better with something like 'at-a-point' vs. 'kriged'.

We chose to keep the term "raw" to emphasize that the isotopic observations were directly sampled without undergoing post-processing or adjustment. This distinguishes them from the kriged values, which involves additional processing. However, we also clarified in the manuscript that the term "raw" also refers to data from a single station only as opposed to an average value from multiple stations.

L162-164: The use of 'time steps' in these sentences is confusing. It might be better to replace the first instance with something like 'finer temporal resolution'.

*We have replaced time steps with a finer temporal resolution at line 156.*

L412: The meaning of "true" model parameterization is unclear.

*By true model parameterization, we refer to the type of SAS function (e.g. PLTI, PLTV or BETATI in our case study) that is best suited to describe the catchment-specific storage and release dynamics. To avoid misunderstandings, we have written best instead of true at line 378.*

L455-456: Not clear in what sense the median transit time has relevant implications for water 'quantity'.

*The implications of median transit time for the water quantity we refer to are water storage, groundwater and hydrological processes, provided in Section 5.3 in the previous version of the manuscript. To make it clearer, we have rephrased this aspect in the revised version in lines 402-418:*

*The value of $TT_{50}$ has relevant implications for both water quantity and quality, as does its uncertainty. The larger the 95% CI in the simulated $TT_{50}$, the greater the difference in the $TT_{50}$ values, which, ultimately, implies distinct hydrological processes, water availability, groundwater recharge and solute export dynamics (McDonnel et al., 2010).*

*For example, knowing the TTD and its uncertainty may be crucial for characterizing the catchment's response to climatic change (Wilusz et al., 2017). Considering the increasing severity of droughts in the past decades (Dai, 2013), a catchment with a shorter $TT_{50}$ and a dominant release of young water might be more affected by droughts than a catchment with a longer $TT_{50}$ whose stream is fed by relatively old water sources. Therefore, a short $TT_{50}$ reveals a low drought resilience of the catchment and limited water availability, which could limit streamflow generation processes and change the instream water quality status during drought periods (Winter et al., 2023). Likewise, TTD uncertainty may affect the understanding of the water infiltration rate, hydrological processes and aquifer recharge, as a shorter $TT_{50}$ suggests that water is quickly routed to the catchment outlet rather than infiltrating deeply into the groundwater. Finally, TTD uncertainty can have an impact on the quantification of the modern groundwater age, i.e., groundwater younger than 50 years (Bethke and Johnson, 2008). According to (Jasechko, 2019) the correct identification of modern groundwater abundance and distribution can help determine its renewal (Le Gal La Salle et al., 2001; Huang et al., 2017), groundwater wells and depths most likely to contain contaminants (Visser et al., 2013; Opazo et al., 2016), and the part of the aquifer flushed more rapidly.*

L489: What does 'data fitting' refer to here?

*Data fitting refers to tracer data interpolation. To be consistent with the rest of the text, we have changed data fitting to tracer data interpolation technique in time and space at line 438.*